# Sexual dimorphism does not translate into foraging or trophic niche partitioning in Peruvian boobies (*Sula variegata*)

Sara Y. Wang[1]*, Carlos Zavalaga[2], Diego Gonzales-DelCarpio[2], Cinthia Irigoin-Lovera[2], Isabella Díaz-Santibañez[2,3], Michael J. Polito[1,4]

1 Department of Oceanography and Coastal Sciences, Louisiana State University, Baton Rouge, Louisiana, United States of America, 2 Unidad de Investigación en Ecología y Conservación de Aves Marinas, Universidad Científica del Sur, Lima, Peru, 3 Institute for Marine and Antarctic Studies, University of Tasmania, Hobart, Australia, 4 Ocean Sciences Department, University of California Santa Cruz, Santa Cruz, California, United States of America

* swang67@lsu.edu

## Abstract

Intraspecific competition can lead to sexual segregation of diets or foraging behaviors in seabirds, and in some species the resulting niche partitioning is facilitated by sexual dimorphism. However, environmental stochasticity can mediate intraspecific competition and thus the extent of sex-based partitioning. The Peruvian booby (*Sula variegata*) is a sexually dimorphic seabird endemic to the Humboldt Current System (HCS), a highly variable environment due to El Niño Southern Oscillation. To determine the extent of sexual partitioning in this species, we quantified the foraging and trophic niches of breeding Peruvian boobies at Isla Guañape Norte, Peru in two years with different oceanographic conditions and nesting propensity. Morphometrics, GPS-tracked foraging behaviors, diets via regurgitates, and isotopic niches were compared between sexes and years where sample sizes permitted. Although females were larger and in better body condition than males, breeding Peruvian boobies in our study did not exhibit sex-specific foraging or isotopic niche partitioning and had few differences in diet. Anchoveta (*Engraulis ringens*) dominated diets in both years, reflecting Peruvian boobies' dependence on this prey. Overall, while oceanographic conditions in 2016 were unfavorable enough to reduce nesting propensity, these effects did not qualitatively translate to foraging or dietary niche partitioning between the sexes for those individuals who opted to breed. In combination, our results suggest weak intraspecific competition during our study period, and highlight how the foraging strategies of Peruvian boobies have adapted to the variable environmental conditions found in the HCS.

## Introduction

Competition occurs between organisms sharing a common resource such as food, with its degree dependent on the scarcity of the resource or the density and abundance of the consumers [1]. When organisms gather in large aggregations – such as dense colonies of breeding seabirds – density-dependent competition is likely to occur, and the competition for limited

**Data availability statement:** The data is available on Dryad at: https://doi.org/10.5061/dryad.kd51c5bdv

**Funding:** Funding was awarded to CZ under grant number FONDECYT 152-2015, "Implementación de Nuevas Técnicas para el Monitoreo Biológico de las Aves Guaneras en el Perú". Funders were Fondo Nacional de Desarrollo Científico y Tecnológico y de Innovación Tecnológica (https://portalanterior.prociencia.gob.pe/) and Universidad Científica del Sur (https://www.cientifica.edu.pe/). None of the funders played any role in the study design, data collection and analysis, decision to publish, or preparation of the manuscript.

**Competing interests:** The authors have declared that no competing interests exist.

prey resources may be intensified further by the constraints of central-place foraging and prey depletion around the colony [2–5]. To alleviate the intensified pressure of this competition, seabirds may switch diet, change feeding and foraging strategies, and/or segregate themselves spatially, temporally, or behaviorally [6,7].

Within a species, niche partitioning can occur between multiple demographic units, such as age, sex, or breeding status [6,8]. Intraspecific segregation in diet or foraging habitat may arise from competitive exclusion and/or habitat or niche specialization due to size-related sexual differences in anatomy or parental role [9]. In seabirds, studies have observed intraspecific sexual segregation in foraging behaviors, dietary niche, or habitat utilization [10–12]. However, sexual dimorphism may not necessarily translate to trophic niche differentiation. In a review of published studies, 70% of sexually dimorphic seabird species exhibited trophic or spatial segregation in temperate and polar regions versus 19% in tropical species [13], suggesting that sexual segregation may be environmentally driven.

Moreover, diets and/or foraging behavior within a population may also vary over time according to environmental conditions and prey abundance [14]. When preferred prey species become scarce, seabirds may respond by opportunistically broadening their isotopic niche width [15], increasing their foraging effort [16–18], or diverging in their diet [19]. Sexual segregation can act to reduce competition for food resources when environmental conditions are poor [20,21]. Such dietary and foraging plasticity can prove useful in the face of environmental stochasticity.

The sexually dimorphic Peruvian booby (*Sula variegata*) [22] breeds in the Humboldt Current System (HCS) and must deal with the demands of an environment with high interannual variability in prey abundance and aggregation [23]. The HCS lies off the west coast of South America and is marked by coastal upwelling, enhanced primary productivity, and an intense and shallow Oxygen Minimum Zone (OMZ) [24,25]. Unusually productive, especially at tropical latitudes, the HCS supports high abundances of both seabirds and forage fish: a bounty of anchoveta (*Engraulis ringens*), which dominates the food web [26], supports the formation of dense colonies of Peruvian boobies during the breeding season. However, the HCS is subject to frequent perturbations and exhibits great oceanographic variability over multiple scales [24]. The Índice Costero El Niño (ICEN), largely focused on localized impacts on the coastal Peruvian region [27], characterizes the interannual fluctuations in sea surface temperature (SST) in the HCS by classifying conditions into three main categories: colder than average (La Niña), warmer than average (El Niño), and neutral. This interannual variation in SST, closely related to upwelling and the depth of the OMZ, can suppress or enhance productivity and anchoveta abundance and accessibility to Peruvian boobies and other predators [28,29], and in some cases lead to species assemblage shifts in the pelagic food web [30].

Historically, studies examining the diet and foraging ecology of Peruvian boobies relied on traditional diet analysis from regurgitates [31], although biogeochemical methods have become more common [32]. Peruvian boobies are known to feed preferentially on anchoveta, with this species comprising up to ~78% of their diet, and consumption is strongly correlated with anchoveta biomass [33]. However, in unfavorable El Niño years, the proportion of anchoveta in the diet shrinks [34], and these years can be marked by high mortality and nesting failure [35]. In the northern boundary of Peru's upwelling system, birds consume almost 100% anchoveta in favorable years, while abandoning islands completely in poor conditions [36]. Stable isotope-based analysis of the HCS estimates that Peruvian boobies occupy a trophic position of 4.5 [37], though the few other studies that have examined stable isotope values in Peruvian boobies have focused on mercury biomagnification [32,38], not diets per se. One prior study found no differences between sexes in either diet composition or the number of anchoveta in regurgitates [26]. To our knowledge, no other studies using regurgitates and/

or stable isotopes have examined sex-based differences in diet composition, prey size, prey mass, or trophic niche.

Past studies have also used biologging techniques to quantify the foraging behaviors of Peruvian boobies at various sites across their breeding range [39–41]. Although Peruvian boobies exhibit reverse sexual dimorphism, in which the female is larger than the male [22,42], only two prior studies have examined foraging trip metrics through the lens of sexual segregation [26,42]. In one study, males foraged slightly farther from the colony and had longer total foraging trip distances than females [42], while in the other, females dove slightly deeper and spent a larger proportion of the foraging time sitting on the water [26]. Despite few sex-specific differences, both studies focused on years with generally similar oceanographic conditions, and thus hypothesized that sex-specific foraging differences may occur during years of lower food availability [26,42].

The goal of our study is to assess sex-based variation in the foraging and trophic niches of Peruvian boobies. Specifically, we sampled breeding birds at Isla Guañape Norte, Peru in 2016 and 2019, in which birds experienced El Niño conditions and neutral ICEN conditions, respectively. We compared morphometrics, foraging behaviors using GPS tracking, diets via regurgitates, and isotopic niches between sexes. We expected moderate reverse sexual dimorphism in Peruvian boobies, as noted by prior studies, and assessed the potential for foraging or dietary niche partitioning between sexes. Where data availability and sample sizes allowed, we also quantitatively compared a subset of these metrics to assess interannual variation in the foraging ecology of Peruvian boobies to explore whether females at Isla Guañape Norte may be more buffered from environmental variation than males due to their larger body size [43].

## Materials and methods

### Study site and ENSO conditions

Isla Guañape Norte (8.545°S, 78.964°W) is an island located approximately 9 km offshore of northern Peru and is the site of a large breeding colony of Peruvian boobies. It is part of a protected natural reserve, La Reserva Nacional Sistema de Islas, Islotes, y Puntas Guaneras (The National Reserve System of Guano Islands, Islets, and Capes) and is administrated by Servicio Nacional de Áreas Naturales Protegidas por el Estado (SERNANP). Peruvian boobies are present on the island year-round, numbering 192,000 individuals in a December 2019 census (Zavalaga, unpublished data). Breeding in Peru is seasonal but asynchronous and takes place primarily in the second half of the year [44]; on Isla Guañape Norte, birds breed between August through March (Zavalaga, unpublished data).

We visited the island during the breeding season in two years: December 2016 and November–December 2019. According to the Índice Costero El Niño (ICEN), birds were experiencing weak El Niño conditions (0.47) in December 2016, and neutral conditions (−0.40) in November 2019 [27,45]. The ICEN is a running 3-month mean sea surface temperature (SST) anomaly for the Niño 1 + 2 region index and was developed by Estudio Nacional del Fenómeno "El Niño" to identify ENSO events that specifically impact coastal Peru [27]. Prior studies of marine birds have used this oceanographic index to reflect the local conditions along these offshore islands of Peru [15].

For context, the weak El Niño conditions in December 2016 were embedded in the middle of a prolonged period of strong to moderate El Niño conditions. Conditions during sampling were preceded by a very strong El Niño (>1.78) from June 2015–January 2016, which gradually weakened to a prolonged moderate-to-weak El Niño that lasted most of 2016 (February–July) [27,45]. The 2016 sampling period was followed by another moderate El Niño (February–April 2017). During the weak El Niño conditions found in December 2016, our 60

x 100 m study plot where breeding adults were captured on Isla Guañape Norte had 45 active nests in 2016. In contrast, in November 2019, this same study plot had approximately 750 active nests, and the neutral conditions during this time were preceded by neutral or La Niña conditions during much of 2018 and 2019, with a brief weak El Niño event from November 2018–April 2019 [27,45]. The 2019 sampling period was followed by an extremely prolonged period of neutral to La Niña conditions (several years post-sampling). Based on this information, we inferred that environmental conditions in December 2016 were generally unfavorable for breeding Peruvian boobies compared to conditions in November 2019.

### Capture and handling of birds

We sampled Peruvian boobies in December 2016 and in November 2019. Breeding adults were captured at the nest soon after dawn (06:00–08:00) using a monofilament lasso attached to a 5-meter telescopic pole. We captured 14 females and 9 males in 2016, and 20 females and 12 males in 2019. Both parents were present at the nest during capture, ensuring that chicks or eggs were not left unattended. Sex of the captured bird was assigned from observed vocalizations at the nest, as whistles are performed only by males, whereas grunts or goose-like honk vocalizations are performed only by females [22].

### GPS tag deployment

We tagged a subset of captured breeding birds with a GPS device, which was attached to the central tail feathers using Tesa® 4651 waterproof tape. In 2016, tracked adults had eggs and/or chicks less than 7 days old while in 2019 tracked adults had chicks between 3–8 weeks old. In total, we successfully tracked 7 females and 3 males in 2016 (15 foraging trips total), and 20 females and 12 males in 2019 (60 trips). Tagged birds had their head feathers marked using PAINTSTIK® livestock markers (LA-CO Industries, Inc.). In 2016, both i-gotU GT 600 (30 grams) and GyPSy-5 (14 grams) GPS devices were used, set to record fixes every 2–10 seconds, and waterproofed by placing each device inside a condom and sealing it inside a heat-sealed polypropylene plastic bag (1 gram). In 2019, Axy-Trek Marine GPS devices weighing 32 grams (Technosmart Europe S.r.l., Rome, Italy) were used to tag birds and set to record GPS locations and dive depth in continuous mode every 1 second. GPS weight was 2% of the weight of the lightest tagged bird (1200 g). Tag masses of 37 or 57 grams (ranging from 2.55–4.75% of body mass), attached to Peruvian boobies using similar methods, were found to have no effect on movement metrics such as trip duration and distance traveled [40]. Birds were recaptured late afternoon (15:00–18:00) on the day of tagging.

The sample size of individuals and foraging trips in 2016 was lower due to the unfavorable environmental conditions and low nesting propensity observed in this year. While the number of trips obtained in 2016 is less than the 25–50 trips per group recommended by other studies [46], the individuals tracked in 2016 actually represent a larger proportion of active breeders than those tracked in 2019. Specifically, in 2016 we obtained tracks from 6.7% and 15.6% of breeding males and females found in the study plot, while in 2019 we obtained tracks from 1.6% of males and 2.7% of females found in the study plot, respectively. While the GPS sample sizes obtained in 2016 are less than ideal, as little is known about the movement and foraging ecology of breeding adults during El Niño events due to their lower nesting propensity these data are inherently valuable and qualitatively interpreted here.

### Morphometric analysis

The weight (g), culmen length (mm), tarsus length (mm), and wing chord (mm, naturally arched) were measured for all but three GPS-tagged individuals. This resulted in a sample size

of 7 females and 3 males in 2016, and 18 females and 11 males in 2019. Given the low sample size of measured males available in 2016, we refrained from statistical analyses of interannual differences and only qualitatively compared morphometric data between years. Otherwise, sex-related differences in weight, culmen, tarsus, and wing chord were statistically analyzed using one-way ANOVAs for all birds pooled across years. To determine the extent of sexual dimorphism, a percent difference size dimorphism index (SDI) was calculated from the mean morphometric measurements of males and females. We used an equation defined in Angel et al. [47] which was based on the methods of Lovich & Gibbons [48]:

$$SDI = \left| -\left( \frac{mean\ male}{mean\ female} \right) + 1 \right| *100$$

We generated a body condition index (BCI) by performing a principal components analysis (PCA) on the culmen, tarsus, and wing chord measurements of all individuals, then regressing the first principal component (PC1, with an eigenvalue of 2.258 explaining 75.3% of variance) of each individual against body mass, and finally assigning the residuals of this regression as the body condition index. Sex-related differences in body condition were also analyzed using a one-way ANOVA. Assumptions of independence and normal distribution were checked using Shapiro-Wilk and Bartlett's tests for all statistical analyses that used morphometric parameters.

## Movement analysis

**GPS data processing.** After retrieval and download of the GPS tracks, the data were recovered from the devices, examined in ArcGIS, and processed prior to analysis. In total, 10 individuals in 2016 and 32 individuals in 2019 yielded usable tracks. As birds were all nesting at the colony, for each track, we first removed all points inside of a 300-meter radius from a point designated as the colony center. We then noted departures and returns for each foraging trip. If a single individual left and returned to the colony center multiple times, each unique segment between a departure/return pair of points, as bounded by start and end time, was defined as separate trips. After confirmation of foraging trips, we then visually checked for GPS data such as non-foraging trip activity (bathing behavior near colony immediately after capture and tagging of the individual) and GPS errors (single outlier points). These data points were removed from tracks either manually or using a similar radius approach.

**Foraging trip metrics.** We calculated total trip distance, maximum distance from colony, and trip duration in using the *sp* [v.1.4-5; [49]], *adehabitatLT* [v.0.3.27; [50]], and *lubridate* [v1.7.9; [51]] packages in R [v4.2.2; [52]]. Specifically, we determined the duration of each trip by calculating the absolute date/time difference between the earliest departure point and the latest return point. We determined maximum distance from colony by specifying the latitude/longitude coordinates of the colony, calculating the distance to colony from each GPS point (km), and taking the maximum value returned for each bird and each trip. We determined total trip distance by calculating the distance between successive points via taking the difference between the latitude/longitude coordinates of each pair of points and totaling up these distances for each individual trip.

**Characterization of at-sea behaviors.** We characterized at-sea behaviors of boobies using the Expectation Maximization binary Clustering (*EMbC*) algorithm [v2.0.4; [53]], an R package which has also been used by others to interpret and analyze movement data from sulid foraging behavior [43,54–56]. The EMbC algorithm is a robust, minimally-supervised multi-variate clustering algorithm [57]. It uses two input variables (speed and turning angle) to determine and assign one of four behaviors to sets of velocity/turn pairs in the movement

data. These four behaviors are defined by the EMbC algorithm as: low velocity/low turning angle (LL), low velocity/high turning angle (LH), high velocity/low turning angle (HL), and high velocity/high turning angle (HH). These assignments can be interpreted as birds that are resting (LL); intensive foraging, also known as intensive searching (LH); traveling (HL); and relocating, also known as extensive searching (HH) [57].

**Kernel density and utilization distribution.** After EMbC behavior assignment, we used the *adehabitatHR* package [v0.4.21; 58] to generate kernel densities and kernel utilization distribution (kernel UD) estimates for GPS points labeled as intensive foraging (LH) [59]. We calculated 50% (core foraging) and 95% (home range) kernel UDs for four groups: 2016 males, 2016 females, 2019 males, and 2019 females [43]. We determined the most appropriate smoothing parameter $h$ using the *ad hoc* method for each group [60]. To obtain intensive foraging areas for 95% and 50% kernels, we created a boundary line following the coastline and clipped each kernel UD by this boundary to exclude foraging areas that lay over land.

The amount of overlap in utilization distributions (UD) were calculated using Bhattacharyya's affinity (BA), which is a statistical measure of affinity between two populations [61,62]. Specifically, we quantified the degree of similarity between population-level UD estimates using the BA kernel UD overlap index, which is a robust estimator of overlap that ranges from 0 (no overlap) to 1 (identical UDs) and assumes the populations use space independently of one another [59,61,62]:

$$BA = \int\limits_{-\infty}^{\infty}\int\limits_{-\infty}^{\infty} \sqrt{\widehat{UD_1}(x,y)} \times \sqrt{\widehat{UD_2}(x,y)}\,dx\,dy$$

Following the methods of Almeida et al. [43], we used a randomization technique to test the null hypothesis that there was no difference in spatial distributions between sexes in 2019. If the null hypothesis is true, overlap between group 50% and 95% kernel UDs should not differ significantly from of that calculated if those groups were randomly assigned. $p$-values were determined by the proportion of random overlaps (out of 1,000 randomizations) that were smaller than the observed overlap. $p$-values < 0.025 or > 0.975 reflect significant differences between comparison groups. We did not apply this null hypothesis testing to the kernel UD data from 2016 due to the low sample sizes of tracked males available in this year.

**Statistical analysis of GPS data.** We first examined the relationships between foraging trip metrics (total trip distance, maximum distance from colony, and trip duration) and at-sea behaviors (proportions of time spent intensive foraging, resting, traveling, and relocating) by creating correlation matrices using Spearman rank correlation. Total distance, maximum distance, and trip duration were all significantly and strongly positively correlated with each other (all $r_{(75)} > 0.92$, $p = 0$). Of the at-sea behaviors, foraging was significantly correlated with traveling ($r_{(75)} = -0.54$, $p < 0.001$) and resting ($r_{(75)} = 0.51$, $p < 0.001$) but not with relocating ($r_{(75)} = -0.54$, $p = 0.310$). Based on these relationships, we proceeded to test only trip duration, proportion of time spent foraging, and proportion of time spent relocating.

Prior to statistical testing, assumptions of normal distribution and independence were tested using Shapiro-Wilk and Levene's tests. Generalized linear mixed models (GLMMs) were performed on data that violated the assumptions: foraging trip metrics (trip duration) and at-sea behaviors (proportions of time spent intensive foraging and relocating). For all models, sex was specified as a fixed effect, and individual bird identity was specified as a random effect. Trip duration was log-transformed before being fitted to the GLMM with a Gaussian distribution; residuals of the model and random factor were checked. In the GLMM for at-sea behaviors, models were weighted by total number of GPS relocations for each bird,

and untransformed data were fitted using a binomial distribution. All movement data analyses were conducted in R [v4.2.2; 52] and RStudio [v.3.1.446; 63], using the *lme4* [v.1.1.35.1; 64] and *car* [v3.1.2; 65] packages. Given the low sample size of tracked males available in 2016, we focused the above statistical analyses solely on data from 2019 and only qualitatively assessed GPS-related data from 2016.

## Regurgitate analysis

Dietary and regurgitate analysis was carried out using the methods of Zavalaga et al. [66]. In 2016, sampling occurred from December 15th to the 22nd, and we induced regurgitation of randomly selected birds when they returned to the nest ($n = 82$). Unfortunately, the sex of birds was not recorded in 2016. In 2019, sampling occurred from November 16th to December 8th, and we induced regurgitation in a subset of sexed birds that were equipped with GPS loggers immediately upon their return from a feeding trip ($n = 25$). This resulted in a sample size of 13 females and 12 males in 2019. After the identification of prey items in regurgitates, we assessed the frequency of occurrence of each prey species, diet composition by species (percent mass), total regurgitate mass, number of prey items, and individual prey size (total length in mm).

The relatively larger sample sizes for regurgitate relative to GPS analysis permitted statistical comparison of these data between sexes in 2019 and between years with sexes combined. Prior to these analyses, we first examined the relationships between individual regurgitate metrics by creating a correlation matrix using Spearman rank correlation. Across all comparisons, only total mass and total number of prey resulted in a moderate correlation ($r_{(83)} = 0.66$, $p < 0.001$). As such we chose to proceed with all statistical tests for all regurgitate variables. Specifically, we used separate, non-parametric Mann-Whitney U tests to analyze the effects of year (2016 vs. 2019) and sex (males vs. females in 2019) on diet composition, total prey mass, and number of prey items, as these data violated assumptions of normality and/or homogeneity in variance based on Shapiro-Wilk and Levene's tests. As each regurgitate contained multiple fish, we calculated the mean prey size for each regurgitate prior to further statistical analysis. While mean prey size data met the assumption of normality, it did not meet the assumption of homogeneity in variance between sexes. Therefore, we used separate Welch's t-tests to analyze the effects of year and sex on mean prey size. Although this test is parametric, it does not assume homogeneity of variance and is more robust to deviations from normality. Lastly, to quantify prey size heterogeneity between sexes, we calculated the coefficient of variation of mean prey size for males and females in 2019. All regurgitate data analyses were conducted in R [v4.2.2; 52] and RStudio [v.3.1.446; 63].

## Stable isotope analysis

**Sample collection.** We collected 0.5 ml of blood from the tarsal vein of all captured birds (i.e., 14 females and 9 males in 2016 and 20 females and 12 males in 2019). We preserved collected blood in vials with 1 ml of 99.9% ethanol from a common source [67]. Prior studies have generally found no effect on the stable isotope values of whole blood via preservation in ethanol [67–69, though see 70]. While isotopic turnover of blood has not been measured in sulids, in other piscivorous seabirds the $\delta^{13}C$ and $\delta^{15}N$ values of whole blood reflects dietary information over the past 20–28 days [71,72].

**Sample preparation.** Peruvian booby whole blood samples were dried and homogenized using a mortar and pestle. Approximately 0.6 mg of each sample was then loaded into tin cups and flash-combusted using a Costech ECS4010 elemental analyzer. These samples were analyzed for carbon and nitrogen stable isotopes ($\delta^{13}C$ and $\delta^{15}N$) using an interfaced

Thermo Delta XP continuous flow stable isotope ratio mass spectrometer. Raw $\delta$ values were normalized on a two-point scale using glutamic acid reference materials with low and high values (i.e., USGS-40 ($\delta^{13}$C = −26.4 ‰, $\delta^{15}$N = −4.5 ‰) and USGS-41 ($\delta^{13}$C = 37.6 ‰, $\delta^{15}$N = 47.6 ‰)). Sample precision based on repeated sample and reference material was 0.1 ‰ and 0.2 ‰ for $\delta^{13}$C and $\delta^{15}$N, respectively. Stable isotope ratios are expressed in $\delta$ notation in per mil units (‰), according to the following equation:

$$\delta^X = \left[\left(\frac{R\ sample}{R\ standard}\right) - 1\right] * 1000$$

where $X$ is $^{13}$C or $^{15}$N, and $R$ is the corresponding ratio of $^{13}$C/$^{12}$C or $^{15}$N/$^{14}$N. The $R_{standard}$ values were based on the Vienna PeeDee Belemnite (VPDB) for $\delta^{13}$C and atmospheric N$^2$ for $\delta^{15}$N.

**Isotopic niche analysis.** Similar to regurgitate analysis, the relatively larger sample sizes of stable isotopes compared to tracking data, especially for males in 2016, allowed for statistical tests for both interannual and sex differences in isotopic niche. We compared the isotopic niche position, width and overlap among the four sex/year groups (males in 2016, females in 2016, males in 2019, and females in 2019). Isotopic metrics such as isotopic niche position are commonly used as proxies for habitat and resource use in consumers (e.g., $\delta^{13}$C can indicate foraging habitat and basal resource use, while $\delta^{15}$N can indicate trophic position) [73,74]. For each group, this was examined using univariate and multivariate analyses following the methods of Hammerschlag-Peyer et al. [75]. Differences in bivariate niche position ($\delta^{13}$C and $\delta^{15}$N) were determined by calculating mean Euclidean distances (*ED*) between the centroid means of each group. Bivariate isotopic niche positions were considered to be different if the Euclidean distance between two groups was significantly greater than zero in comparison to a null distribution generated by a residual permutation procedure [76]. Differences in univariate niche position ($\delta^{13}$C or $\delta^{15}$N) were then identified using two-way ANOVAs (with factors as sex and year and their interaction). Tukey's tests were also performed for $\delta^{13}$C and $\delta^{15}$N group means.

Isotopic niche width is a proxy for the variety of resources consumed by consumers and can be calculated using both Frequentist [75] and Bayesian [77] methods. Using the Frequentist framework, we calculated the bivariate niche widths of each group using the mean distance to centroid (*MDC*), which is the average Euclidean distance from each individual sample to the centroid mean of its group [75,76]. Bivariate niche widths were considered to be different if the absolute value of *MDC* differences between two groups were greater than zero in comparison to a null distribution generated by a residual permutation procedure [76].

Using the Bayesian framework, we calculated the bivariate niche widths of each group using standard ellipse areas (*SEA$_b$*) in the Stable Isotope Bayesian Ellipses in R (*SIBER*) package [v2.1.9; 78]. Bayesian standard ellipses are calculated using Markov Chain Monte Carlo simulations and describe the mean bivariate dispersion of a group [77]. SEA$_b$ captures the same properties as standard ellipse area, but is unbiased with respect to sample sizes as low as 8–10 per group [77]. *SEA$_b$* was calculated encompassing 95% of data points to maintain consistency between Bayesian niche width and niche overlap calculations. We ran 2 chains of 20,000 iterations, with a burn-in of 1000 and thinning of 10. We compared *SEA$_b$* between groups using their posterior probabilities (*PP*), setting *PP* > 0.95 to identify significant differences. We calculated *PP* as the probability that the posterior *SEA$_b$* of one group was different from that of another group in a pairwise comparison.

When differences in bivariate niche width were identified using Frequentist or Bayesian approaches, we then tested for differences in univariate niche width ($\delta^{13}$C or $\delta^{15}$N) using Bartlett's test to test for homogeneity in variance. This allowed us to determine whether carbon and/or nitrogen were driving the difference in niche width between groups [75].

Isotopic niche overlap calculations can provide an extra dimension of information to assess the extent of partitioning or similarity in resource use between groups of interest [79]. We examined isotopic niche overlap using the *nicheROVER* package [v1.1.2; 80] in R to calculate the probability of one group's niche area (*SEAb*) falling within the niche area of another group. *SEAb* used in the *nicheROVER* analysis encompassed 95% of the data points within each group and were fed into a Bayesian model with Markov Chain Monte Carlo simulations. We ran 2 chains of 20,000 iterations, with a burn-in of 1000 and thinning of 10. The resulting isotopic niche overlap values (%) are presented as pairwise, median overlap values with 95% upper and lower credible intervals.

Isotopic data were tested for normality with a Shapiro-Wilk test prior to analysis. Isotopic values were recorded as means and standard deviations. All isotopic data analyses were conducted in R [v4.2.2; 52] and RStudio [v.3.1.446; 63].

## Results

### Morphometrics

Body mass ($F_{1,37} = 87.36$, $p < 0.001$), culmen length ($F_{1,37} = 32.53$, $p < 0.001$), tarsus length ($F_{1,37} = 20.4$, $p < 0.001$), and wing chord ($F_{1,37} = 25.09$, $p < 0.001$) differed between sexes. Females were larger than males for all morphometric measurements (Table 1). The size dimorphism index indicated that females had 17.4% larger body mass than males, 6.0% longer culmen, 5.4% longer tarsus, and 3.4% longer wing chord. Body condition differed significantly between sexes ($F_{1,37} = 7.82$, $p = 0.008$). Specifically, in 2019, males on average had lower body condition than females (Table 1). While not statistically analyzed due to the low sample sizes, males qualitatively appeared to also have lower body condition than females in 2016 (Table 1). In addition, morphometric measurements and body condition appeared qualitatively similar between 2016 and 2019.

### Foraging movements

**Foraging trip metrics.** Sex did not have a significant effect on the trip duration of Peruvian boobies in 2019 (Table 2; $F_{1,30} = 0.55$, $p = 0.466$). While not statistically analyzed due to the low sample sizes, trip durations appeared qualitatively similar between sexes in 2016 and between the two years of our study (Table 1).

**Table 1. Morphometric measurements of male and female Peruvian boobies.**

| Year | Sex | Individuals (n) | Mass (g) | Culmen (mm) | Tarsus (mm) | Wing chord (mm) | BCI |
|------|-----|-----------------|----------|-------------|-------------|-----------------|-----|
| **2016** | Female | 7 | 1537 ± 105 (1465–1740) | 96.1 ± 3.4 (92–102) | 54.3 ± 1.8 (50.9–56.1) | 407 ± 9 (393–418) | 18.6 ± 83.9 (−92.2–161.0) |
| | Male | 3 | 1252 ± 34 (1225–1290) | 89.0 ± 6.6 (81.4–93.3) | 49.8 ± 1.7 (48.2–51.5) | 398 ± 14 (388–414) | −53.9 ± 122.0 (−134.0–86.1) |
| **2019** | Female | 18 | 1544 ± 86 (1400–1700) | 96.4 ± 3.0 (91.5–102) | 53.0 ± 2.2 (48.8–57.8) | 410 ± 8 (392–425) | 33.5 ± 89.4 (−142–200.0) |
| | Male | 11 | 1280 ± 89 (1100–1450) | 90.9 ± 1.8 (88.4–93.1) | 50.7 ± 1.5 (48.4–53.9) | 394 ± 8 (382–410) | −52.0 ± 86.6 (−216.0–97.1) |
| **All years** | Female | 25 | 1542 ± 89 (1400–1740) | 96.3 ± 3.0 (91.5–102) | 53.4 ± 2.1 (48.8–57.8) | 409 ± 8 (392–425) | 29.4 ± 86.4 (−142.0–200.0) |
| | Male | 14 | 1274 ± 80 (1100–1450) | 90.5 ± 3.1 (81.4–93.3) | 50.5 ± 1.5 (48.2–53.9) | 395 ± 9 (382–414) | −52.4 ± 89.7 (−216–97.1) |

Mean body mass (g), morphometrics (mm), body condition index (BCI) ± standard deviation and range for male and female Peruvian boobies measured on Guañape Norte, Peru by year. In both years for all measurements, females were significantly larger and in better body condition than males.

**Table 2. Foraging trip metrics for male and female Peruvian boobies.**

| Metric | 2016 | | 2019 | |
|---|---|---|---|---|
| | Female | Male | Female | Male |
| Individuals (n) | 7 | 3 | 20 | 12 |
| Total Trips (n) | 11 | 4 | 41 | 19 |
| Trip Duration (hours) | 2.2 ± 2.0 (0.6–6.3) | 3.2 ± 2.6 (1.0–6.4) | 1.7 ± 1.3 (0.2–7.3) | 1.7 ± 0.8 (0.6–3.4) |
| Trip Distance (km) | 89.4 ± 63.6 (24.8–206.6) | 113.2 ± 80.5 (36.8–198.7) | 72.9 ± 49.7 (7.4–266.7) | 75.2 ± 39.7 (22.1–150.4) |
| Maximum Distance from Colony (km) | 35.9 ± 26.8 (7.2–86.6) | 45.5 ± 31.2 (14.2–74.4) | 24.6 ± 14.2 (4.1–61.3) | 28.1 ± 17.1 (9.5–63.6) |
| Proportion Foraging (%) | 8.5 ± 7.0 (0.3–26.3) | 8.6 ± 6.1 (2.8–17.3) | 5.2 ± 3.0 (1.7–11.6) | 5.2 ± 4.0 (1.1–18.6) |
| Proportion Resting (%) | 32.4 ± 25.1 (1.4–67.3) | 22.2 ± 26.6 (5.4–61.8) | 35.9 ± 35.7 (1.4–95.0) | 16.5 ± 17.1 (0.4–68.1) |
| Proportion Traveling (%) | 37.0 ± 31.9 (0–91.4) | 51.1 ± 33.9 (1.1–74.0) | 35.3 ± 38.9 (0–94.5) | 27.1 ± 36.5 (0–86.9) |
| Proportion Relocating (%) | 22.0 ± 10.0 (6.9–41.3) | 18.1 ± 3.6 (13.5–21.9) | 34.1 ± 35.5 (0–94.5) | 51.1 ± 34.4 (4.4–91.5) |
| 95% (50%) Kernel Area (km²) | 12,846 (2,159) | 2,036 (223) | 3,586 (589) | 3,313 (583) |

Mean foraging trip and at-sea behaviors metrics ± standard deviation and kernel areas of male and female Peruvian boobies GPS tracked on Isla Guañape Norte, Peru by year. Range in parentheses.

**At-sea behaviors.** There was no effect of sex on the proportion of time that Peruvian boobies spent foraging ($z = -0.61$, $p = 0.542$) or relocating ($z = 1.00$, $p = 0.317$) in 2019 (Table 2). This trend (i.e., lack of sex differences) was also qualitatively apparent in 2016. Moreover, individuals of both sexes qualitatively appeared to spend proportionally less time foraging in 2019 (5.2% ± 3.4, range: 1.1–18.6%) than in 2016 (8.5% ± 6.5, range: 0.3–26.3%), though this trend could not be confirmed statistically due to the lower sample size of tracked individuals in 2016.

**Foraging area size and overlap.** In 2016, the 95% and 50% kernel areas associated with intensive foraging were 12,846 km² and 2,159 km² for female and 2,036 km² and 223 km² for male Peruvian boobies, respectively. In 2019, the 95% and 50% kernel areas associated with intensive foraging were 3,586 km² and 589 km² for female and 3,313 km² and 583 km² for male Peruvian boobies, respectively. The observed BA overlap of 95% and 50% kernel areas associated with intensive foraging were 0.13 and 0.01 between sexes in 2019 (Fig 1). The observed spatial overlaps between sexes in 2019, while low, are not lower than randomly expected based on mean ± SD of permuted overlaps ($BA_{0.95} = 0.19 \pm 0.18$, $pp = 0.59$; $BA_{0.50} = 0.04 \pm 0.06$, $pp = 0.917$). Specifically, based on the proportion of randomized overlaps that were smaller than the observed overlap ($pp$), the degree of observed foraging area overlap between sexes were either no different or greater than what would be expected by random chance, indicating a lack of spatial partitioning in 2019. The observed BA overlap of 95% and 50% kernel areas associated with intensive foraging was 0.33 and 0.04 between sexes in 2016, 0.14 and 0.01 between years for females, and 0.24 and 0.06 between years for males, although we did not test for significant differences in these overlap metrics.

## Diet

Anchoveta was found in all regurgitate samples across all years and sexes. 100% of samples from both 2016 ($n = 82$) and 2019 ($n = 25$) contained anchoveta. Likewise, for females ($n = 13$) and males in 2019 ($n = 12$), 100% of samples contained anchoveta. Only in 2016

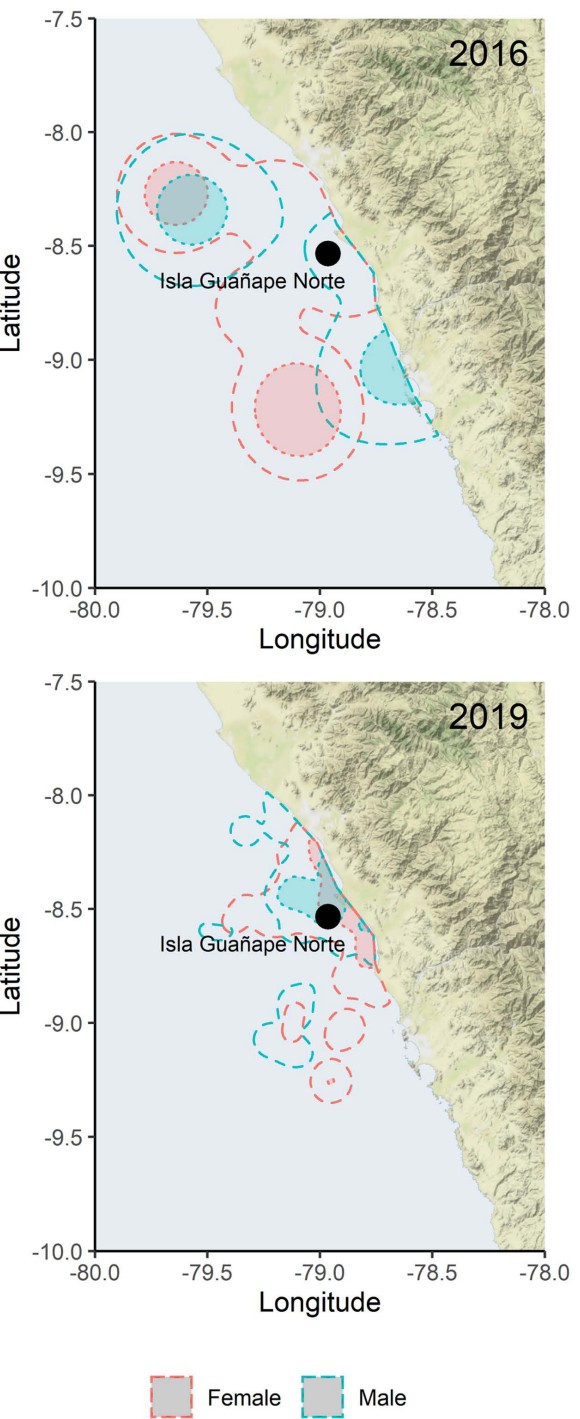

**Fig 1. Kernel densities of Peruvian boobies.** Kernel utilization distributions (UDs) showing intensive foraging areas of female (red) and male (blue) Peruvian boobies GPS tracked on Isla Guañape Norte, Peru in two years, 2016 and 2019. 95% kernels are outlined with a dotted line and core 50% kernels are shaded solid. Map tiles by Stamen Design under a Creative Commons Attribution (CC BY 4.0) license. Data by OpenStreetMap licensed under the Open Data Commons Open Database License (ODbL).

were other prey species found in booby diets: Peruvian silverside (*Odontesthes regia*) was found in 2% of samples (two occurrences) and king gar (*Scomberesox saurus scombroides*) in 1% of samples (one occurrence). Even so, the proportion of diet that was anchoveta by mass did not differ between years ($W = 129.5$, $p = 0.341$). In 2019, the total prey mass in regurgitates averaged 73 g with a standard deviation of ± 41 g (median: 61 g, range: 10–219 g) and 95 ± 63 g (median: 82 g, range: 17–252 g) in 2019, and did not significantly differ between years ($W = 847.5$, $p = 0.193$). However, in 2019, females averaged 127 ± 63 g (median: 118 g, range: 48–252 g) while males averaged 60 ± 43 g (median: 45 g, range: 17–158 g). There was a significant difference in total prey mass between the sexes, with females on average returning with double the prey mass of males' ($W = 129.5$, $p = 0.006$).

The number of prey items in regurgitates averaged 6.6 ± 3.0 items (median: 5.5, range: 1–15) in 2016, while in 2019 it averaged 8.1 ± 5.9 items (median: 7, range: 1–29), and did not significantly differ between years ($W = 94$, $p = 0.396$). In 2019, females averaged 8.3 ± 3.4 items (median: 8, range: 3–16)), while males averaged 7.9 ± 7.9 items (median: 5.5, range: 1–29), and did not significantly differ between sexes ($W = 890$, $p = 0.319$).

Anchoveta prey size averaged 13.3 ± 1.1 cm (median: 13.4 cm, range: 10–15.2 cm) in 2016 and 12.5 ± 2.3 cm (median: 13.4 cm, range: 6.2–14.6 cm) in 2019, and did not significantly differ between years ($F_{25.882} = 1.66$, $p = 0.109$). In 2019, prey size for females averaged 13.6 ± 0.6 cm (median: 13.4 cm, range: 12.7–14.6 cm) while males' averaged 11.6 ± 2.8 cm (median: 12.2 cm, range: 6.1–14.4 cm). Females' regurgitates contained significantly larger prey than males' ($F_{11.954} = 2.47$, $p = 0.030$). The coefficient of variation was 4.4% for females and 24.1% for males, and variances differed significantly between sexes ($F_{1,21} = 13.78$, $p < 0.001$), as males' prey size was more dispersed than in females ([Fig 2]).

## Stable isotopes

**Isotopic niche position.** Males and females occupied similar bivariate isotopic niche positions in both 2016 ($ED = 0.09$, $p = 0.786$) and 2019 ($ED = 0.13$, $p = 0.465$). However, bivariate isotopic niche position differed between years for both male ($ED = 0.66$, $p = 0.001$) and female ($ED = 0.74$, $p = 0.001$) Peruvian boobies ([Table 3]). Univariate analyses indicated that differences in bivariate isotopic niche position were driven by differences in both $\delta^{13}C$ and $\delta^{15}N$ blood values. Blood $\delta^{13}C$ values significantly differed between years ($F_{1,51} = 7.70$, $p = 0.008$). However, sex ($F_{1,51} = 0.98$, $p = 0.327$) and the interaction between sex and year ($F_{1,51} = 0.004$, $p = 0.949$) were not significant for blood $\delta^{13}C$ values. Specifically, blood $\delta^{13}C$ values for both males and females were lower in 2016 (−15.1 ± 0.4 ‰) relative to 2019 (−14.8 ± 0.4 ‰). Similarly, $\delta^{15}N$ values differed by year ($F_{1,51} = 105.56$, $p < 0.001$), but not by sex ($F_{1,51} = 0.74$, $p = 0.394$) or the interaction between sex and year ($F_{1,51} = 0.39$, $p = 0.537$). Blood $\delta^{15}N$ values were higher in 2016 (12.6 ± 0.3 ‰) than in 2019 (12.0 ± 0.2 ‰).

**Niche area.** The Frequentist measure of bivariate isotopic niche width (*MDC*) indicated no significant differences among year and sex combinations (all comparisons $p > 0.122$). The Bayesian measure of bivariate isotopic niche width ($SEA_b$) indicated that 2016 females had a larger $SEA_b$ than 2019 females ($PP = 0.967$); however, all other year and sex combinations did not differ (all $PP < 0.95$). However, univariate measures of niche width did not differ for either blood $\delta^{13}C$ ($X^2$ (3, $N = 55$) = 1.18, $p = 0.758$) or $\delta^{15}N$ values ($X^2$ (3, $N = 55$) = 2.40, $p = 0.494$).

**Niche overlap.** The isotopic niche of females in 2016 overlapped 88.7% (95% CI = 65.3–99.3%) with the isotopic niche of males in the same year. Males in 2016 overlapped 83.8% (95% CI = 53.1–98.6%) with females in 2016. In 2019, the isotopic niche of females overlapped 71.2% (95% CI = 46.7–91.4%) with males. Males in 2019 overlapped 90.5% (95% CI = 66.8–99.1%) with females in 2019.

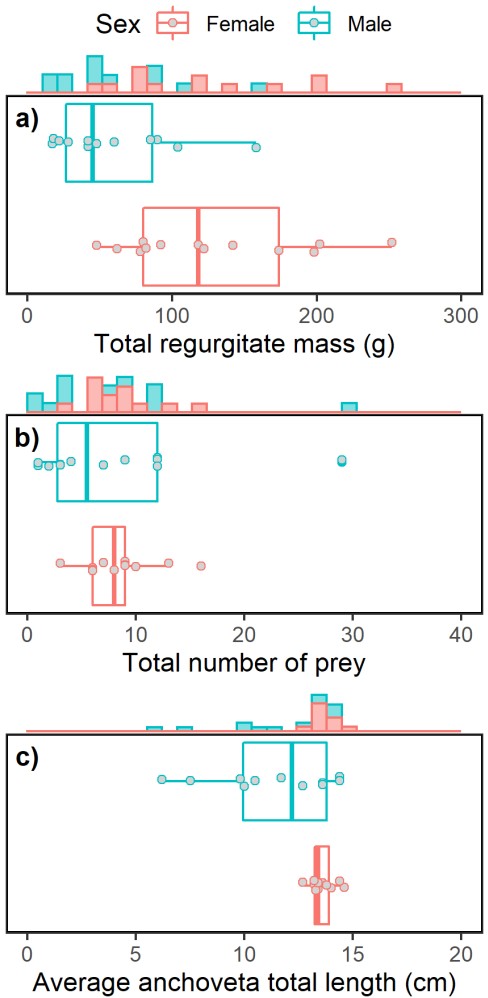

**Fig 2. Prey item measurements of regurgitates of Peruvian boobies.** Total regurgitate mass (g), number of prey items, and average total length (cm) of anchoveta in regurgitates sampled from male and female Peruvian boobies on Isla Guañape Norte, Peru in 2019.

**Table 3. Bivariate isotopic niche positions of male and female Peruvian boobies.**

| Year | Sex | n | δ¹³C (‰) | δ¹⁵N (‰) | C/N | MDC (‰) | SEAb (‰²) |
|------|-----|---|----------|----------|-----|---------|-----------|
| **2016** | Female | 14 | −15.0 ± 0.4 | 12.6 ± 0.2 | 3.29 | 0.44 | 0.43 (0.17–0.72) |
| | Male | 9 | −15.1 ± 0.4 | 12.6 ± 0.3 | 3.23 | 0.45 | 0.35 (0.18–0.55) |
| **2019** | Female | 20 | −14.7 ± 0.4 | 12.0 ± 0.2 | 3.33 | 0.37 | 0.19 (0.09–0.30) |
| | Male | 12 | −14.8 ± 0.3 | 12.0 ± 0.2 | 3.35 | 0.29 | 0.26 (0.15–0.38) |

$\delta^{13}C$ and $\delta^{15}N$ (mean ± standard deviation), carbon/nitrogen ratio, mean distance to centroid (MDC), and Bayesian standard ellipse areas (SEA$_b$) of male and female Peruvian boobies sampled on Isla Guañape Norte, Peru, by year. $\delta^{13}C$, $\delta^{15}N$, MDC, and SEA$_b$ are reported in per mil (‰). SEA$_b$ values are reported with 95% confidence intervals.

The isotopic niche of female boobies in 2016 overlapped 11.2% (95% CI = 1.3–40.8%) with the isotopic niche of female boobies in 2019 (Fig 3). Similarly, females in 2019 overlapped 16.3% (95% CI = 1.5–63.1%) with females in 2016. The isotopic niche of male boobies in 2016 overlapped 16.1% (95% CI = 1.8–48.0%) with the isotopic niche of male

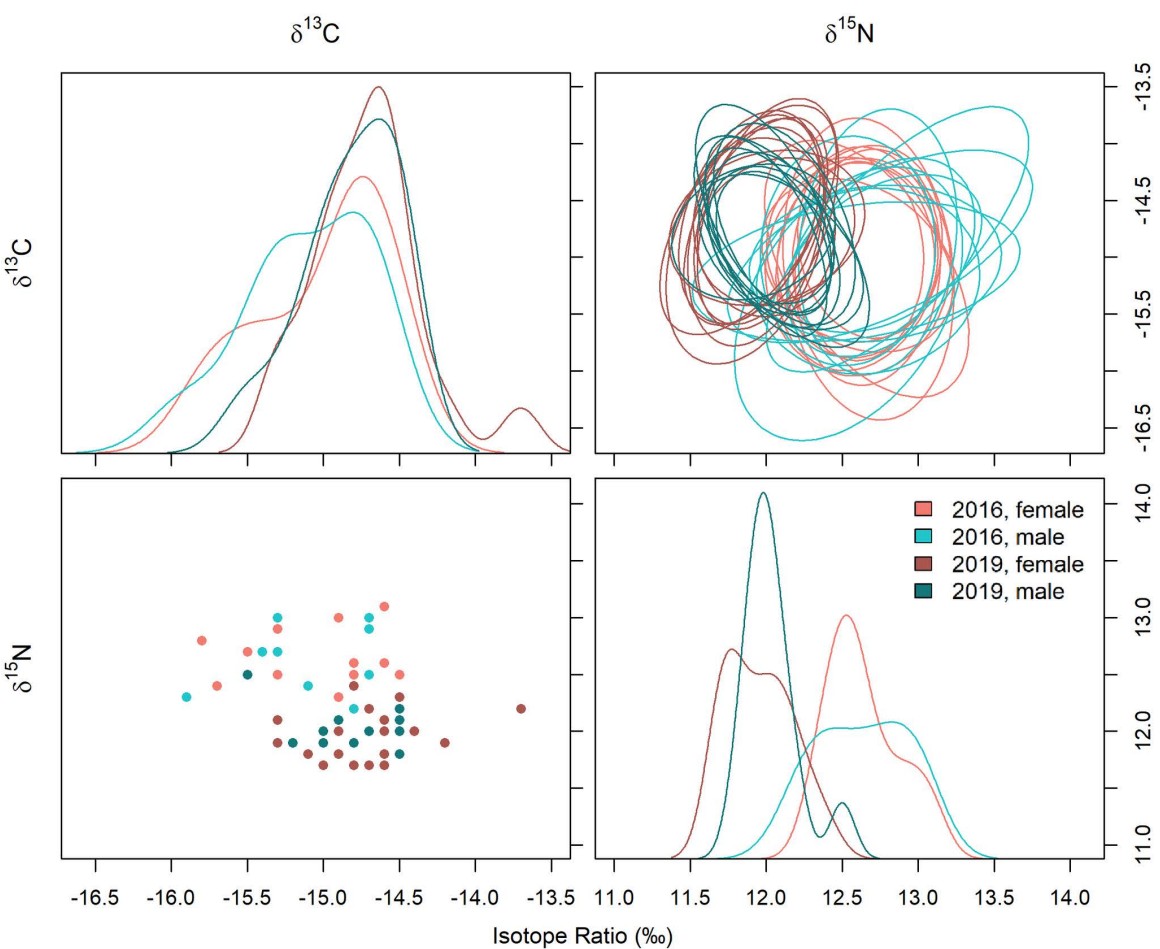

**Fig 3. Isotopic niches ($\delta^{13}$C and $\delta^{15}$N) of male and female Peruvian boobies.** (Top right, bottom left) Isotopic niche space and distribution of carbon and nitrogen isotope values and (top left, bottom right) SEA$_b$ probability distributions and overlap probability of male and female Peruvian boobies on Isla Guañape Norte in 2016 and 2019.

boobies in 2019. Similarly, males in 2019 overlapped 38.65 (95% CI = 4.9–93.0%) with males in 2016.

## Discussion

We studied Peruvian boobies on Isla Guañape Norte, Peru to quantify sex-based variation in their foraging behavior and trophic niches. Although females were moderately larger than males and in better body condition, there was little evidence of sex-specific foraging or dietary niche partitioning. While foraging trip durations did not differ between sexes, individuals appeared to spend proportionally more time foraging during trips in 2016 than in 2019. However, the relatively low sample size of tracked individuals in 2016 relative to 2019 prevented a rigorous statistical confirmation of this trend. Anchoveta was the dominant prey species consumed by Peruvian boobies on Isla Guañape Norte, with few differences in diet metrics between sexes or years. One notable exception was that females captured larger and less variably sized anchoveta than males and had larger regurgitate masses on average. While the isotopic niche of Peruvian boobies did not differ between sexes, they did differ slightly between years. Although sample sizes were limited in some comparisons, our results suggest a general

lack of significant foraging or dietary niche partitioning between male and female Peruvian boobies on Isla Guañape Norte, Peru for those individuals who opted to breed during the two years of our study.

## Sexual dimorphism and body condition

Females were significantly larger than males in all morphometric measurements. Female Peruvian boobies in our study were 17.4% heavier than males, similar to the 19% reported in a previous study [22]. We also found that males had significantly lower body condition than females during our study. Even so, body condition was qualitatively similar between years despite the lower nesting propensity in 2016 versus 2019. This result may reflect a selection bias, as presumably only adults in good enough condition would opt to breed, especially in an unfavorable year like 2016.

Lower body condition in males relative to females is a pattern that has been reported in other sulid species [81–83]. The observed differences in body condition between sexes in sulid species may be due to their differing sex-based parental roles, or an effect of higher energetic flight costs, lower foraging efficiency, and/or a higher rate of decline of body condition in males due to smaller body size [81,82,84]. For example, a study on red-footed boobies (*Sula sula*) found that female body condition stayed unchanged throughout breeding, while the body condition of males was lower during chick-rearing compared to incubation [82]. If this is the case for Peruvian boobies, our measurements of male body condition may have been biased by measuring a combination of incubating and chick-rearing birds in 2016 and only chick-rearing birds in 2019. However, body condition (i.e., endogenous energy stores) may not solely or necessarily reflect environmental or foraging conditions [85]. Birds may also be choosing to maintain less fat/energy stores to reduce flight costs [85] or to improve dive performance [86], which would become increasingly important over the season as energy expenditure (e.g., from provisioning demands) increases [87].

## Sex-based partitioning

**Foraging by sex.** Body and wing morphology plays a functional role in flight performance [88,89], and differences in such due to sexual dimorphism may lead to differences in foraging. Despite moderate sexual dimorphism and differences in body condition, we found little evidence of spatial or behavioral foraging niche partitioning between male and female Peruvian boobies during our study. Specifically, males and females did not differ in foraging trip duration, though females had larger foraging areas than males in 2019.

Our results therefore corroborate the findings of other studies which reported limited evidence of foraging partitioning between male and female Peruvian boobies. These prior studies have reported similar trip distances and durations between sexes and moderate to high overlap (20–74%) of foraging areas [26,42]. However, these studies do report some differences between sexes. Weimerskirch et al. [42] found that male Peruvian boobies had slightly greater maximum foraging distance than females, and both Zavalaga et al. [26] and Weimerskirch et al. [42] reported deeper dive depths for females. Differences among our study and others may be influenced by a combination of differences in timing of each study relative to the breeding cycle, colony size, and geographic location, and/or local oceanographic conditions and associated anchoveta accessibility, abundance, and condition, all of which can affect foraging behaviors [90].

The presence and degree of sexual differences in foraging varies across sulids [8,9,43,81,91,92]. In studies that have reported sexual differences, female sulids foraged for

longer durations, traveled greater distances, and ranged further from the colony [9,81,91,93]. In addition, brown boobies (*Sula leucogaster*) and red-footed boobies exhibit low sex-based spatial partitioning with sexual segregation dependent on environmental conditions and presence of competitors [43,81]. For example, when red-footed boobies inhabited the same island, male brown boobies shifted their foraging areas while females maintained their previous foraging areas [43].

However, it is important to note that the low sample sizes of individuals and trips in 2016 limit our ability to conclusively confirm the lack of spatial partitioning between sexes in this year. Considering the asymptotic relationship between the number of trips recorded and the representativeness of the resulting data [46], the small number of tracked males in 2016 likely does not fully represent spatial patterns at the population level. While we were not able to quantitatively assess foraging trip representativeness at the population level, the low nesting propensity observed in 2016 meant that the lower number of birds sampled in 2016 relative to 2019 represented a larger proportion of the breeding population in 2016 relative to 2019. Moreover, the lack of differences between sexes in trip duration and space use is not fully unexpected with the constraint of central place foraging and the fact that Peruvian boobies take shorter foraging trips (both in duration and in distance from colony) than other sulids [94]. Future studies with increased sample sizes would be beneficial to confirm our findings, though low breeding propensity during poor conditions will make the goal of tracking a higher number of individuals and trips challenging.

**Diets by sex.** The evidence for dietary differences between male and female Peruvian boobies in our study was mixed. In 2019, both males and females consumed primarily anchoveta and captured similar numbers of prey. However, females had larger regurgitate masses, roughly double that of males, and on average captured larger anchoveta. Females captured a narrower size range of larger anchoveta, while males captured a wider size range of anchoveta that were smaller on average (Fig 2). Unfortunately, regurgitate data was identified to sex only in 2019, so it was not possible to assess if and how these patterns may have been similar or different during the weak El Niño conditions present in 2016.

Only one other study has examined Peruvian booby diets relative to sex using regurgitates. Zavalaga et al. [26] reported that both sexes feed exclusively on anchoveta and do not differ in the number of prey per regurgitation, but did not assess regurgitate mass or prey size differences between sexes. In contrast to foraging behaviors, we may be able to attribute the dietary differences we observed to the species' sexual dimorphism. The differences in prey size and regurgitate mass may be due in part to the larger mass of females enabling them to swallow larger fish and/or carry larger loads, and/or a difference in parental roles where the females are the greater provisioners of chicks [84]. Prey size partitioning between sexes may also be mediated by diving ability in Peruvian boobies, as larger females have been found to dive deeper than smaller males [26,42]. For example, body size is positively correlated to dive depth and prey size in blue-footed boobies from northern Peru, indicating that prey size partitioning can be mediated by sexual size dimorphism in sulids [66].

**Isotopic niche by sex.** The isotopic niche of Peruvian boobies did not differ between sexes in either year of our study, with males and females having similar blood $\delta^{13}C$ and $\delta^{15}N$ values. While prior studies have examined stable isotope values in the tissues of Peruvian boobies, to our knowledge we are the first study to compare the isotopic niches between sexes. The lack of sex-based isotopic niche segregation agrees with the broadly similar and anchoveta-dominated diets in the regurgitates of both sexes in our study. Studies in other sulids have also found that males and females tend to occupy similar isotopic niches. For example, studies have reported similar tissue $\delta^{13}C$ and $\delta^{15}N$ values between sexes in red-footed [8,95], masked [8], brown [9,95], and blue-footed boobies [9].

Some prior studies have reported sex-based differences in stable isotope values. Male brown boobies had lower $\delta^{13}C$ and $\delta^{15}N$ than females on Palmyra Atoll [8] and $\delta^{13}C$ values were higher in female red-footed boobies on Europa Island [91], indicating that females of both species foraged less on pelagic prey and that female brown boobies consumed higher trophic level prey. The reverse pattern was found, however, in breeding brown boobies on Raso Islet, Cabo Verde, where females had significantly lower plasma $\delta^{13}C$ values than males [43]. Cape gannets exhibited environmentally-driven niche partitioning with clear foraging segregation in years of lower food availability, when females took longer foraging trips, had a larger isotopic niche, and fed at a lower trophic level than males [20]. However, we found that male and female Peruvian boobies did not differ in isotopic niche in both years of our study. It is possible that while the 2016 El Niño conditions were strong enough to lower nesting propensity, these lower breeding numbers reduced intraspecific competition such that prey availability was sufficient to preclude the need for sex-based niche-partitioning between the smaller number of pairs that opted to breed. However, it also seems quite possible that if El Niño conditions were more severe it would simply cause the remaining birds to abandon breeding efforts rather than adopt sex-based partitioning. Even so, Peruvian boobies might exhibit environmentally or temporally driven isotopic niche segregation between sexes during the non-breeding season when no longer constrained to central place foraging and the energetic demands of breeding. Future studies examining feather stable isotopes are needed to assess this possibility.

## Interannual variation

**Foraging by year.** Given the low sample sizes of GPS-tracked individuals in 2016 and the slight shift in breeding phenology between years, we exercise caution in our interpretation of observed interannual trends in foraging metrics and behaviors. While conclusively teasing apart the relative influences of environmental and phenological variation on our study results is not possible, foraging metrics such as trip duration appeared qualitatively similar between years. However, our results suggest that individuals may have spent proportionally more of their time foraging in 2016 than in 2019. Moreover, while males had similarly sized foraging areas in each year, females had notably larger kernel areas in 2016 (95%: 12,846 km²) when compared to 2019 (95%: 3,586 km²).

Environmental conditions, differing parental roles between males and females, and breeding phenology may have contributed to Peruvian boobies at Isla Guañape Norte spending qualitatively more time foraging during trips away from the colony in 2016 relative to 2019. For example, the increased time spent foraging may have been influenced by the weak El Niño ICEN conditions in 2016, which is collinear to chlorophyll-$a$ and oxycline depth [38]. A combination of decreased primary productivity and deepened oxycline depth may have reduced anchoveta availability [30], though not to a degree to which would trigger total abandonment of breeding altogether.

Female boobies may have absorbed more of the load of increased foraging effort during the weak El Niño conditions, as seen in the larger intensive foraging kernel areas in 2016 compared to 2019. As anchoveta distribution becomes more sparse, both vertically and horizontally, due to a deepened oxycline [30,96], it may be more efficient for the female partner to take on more of the provisioning duties, as they are able to fly longer distances more efficiently and dive deeper due to a heavier body mass [26]. Peruvian booby males have been noted to take on more of the nest-guarding and brooding duties, spending less time at sea and primarily foraging in shallow waters inshore, while females are greater provisioners of chicks, especially in the later stages of chick growth [94,97]. A similar division of parental duties is seen in other sulids (red- and blue-footed boobies) that exhibit sexual dimorphism [84,98].

While we found little evidence of sex-based foraging partitioning occurring in this species, Peruvian boobies at Isla Guañape Norte may simply have enhanced parental roles in years of lower prey availability.

It is important to restate that in 2016, we tracked both incubating and chick-rearing birds, while in 2019, only chick-rearing birds were tracked. Incubating seabirds generally have longer foraging trip durations than chick-rearing seabirds [99,100], but in a post-hoc analysis of incubating versus chick-rearing birds in 2016, we found no differences between the two groups in trip duration, total trip distance, maximum distance from colony, or proportion of time spent foraging (S1 Table). However, as sample sizes were small, we cannot rule out the possibility that breeding phenology had a confounding effect on comparisons of foraging trip metrics between years. Lastly, selection bias may also have been a contributing factor for the lack of interannual differences observed. Adults that opted to breed in 2016 may reflect a subset of higher-fitness individuals who were able to meet the energetic requirements to rear a brood, and who may be less likely to have to adjust their foraging behaviors and/or diets to do so across years.

Other studies that have sought to examine interannual or seasonal variation in the foraging behaviors of Peruvian boobies are scarce and often similarly limited in sample sizes. One prior study deployed data loggers on three adult chick-rearing birds total in 2000 and 2004 on Isla Pajaros, Chile, and reported similar trip durations between years [41]. Clark et al. [40] tracked breeding Peruvian boobies from Isla Macabi, Peru (located 100 km north of Guañape Norte) in December 2020 and May 2021 and reported differences in maximum and total foraging distances among these two seasons during neutral ICEN conditions [45]. Similarly, Zavalaga et al. [26,90] tracked breeding Peruvian boobies from Isla Lobos de Tierra and Isla Lobos de Afuera, Peru in December 2006 and December 2007 during weak El Niño and moderate La Niña ICEN conditions, respectively [45]. They reported longer trip durations and higher diving frequencies at Isla Lobos de Tierra in 2006 than Isla Lobos de Afuera in 2007 [26,90], but given the study design, were unable to disentangle the possible effect of colony location (inshore vs. offshore) from oceanographic conditions (El Niño vs. La Niña). Unfortunately, the differences in timing, location, and oceanographic conditions found in our study and others make it challenging to compare interannual trends in the foraging behaviors of Peruvian boobies across studies.

**Diets by year.** Peruvian boobies at Isla Guañape Norte did not differ in their regurgitate masses, prey numbers, or prey sizes during the weak El Niño conditions of 2016 relative to the neutral conditions of 2019. Similar regurgitate masses and numbers of prey in two years with differing oceanographic conditions might be due to the observed qualitative increase in foraging effort (i.e., proportion of time foraging) in 2016 relative to 2019. In addition, while anchoveta dominated Peruvian booby diets in both years, other prey species (e.g., Peruvian silverside, king gar) were found in regurgitates in 2016, though with very low frequency of occurrence. Moreover, provisioned chicks generally require larger and more frequent feedings as they grow [4], so smaller regurgitate masses and/or prey numbers might have been expected during 2016 when sampled adults were incubating or feeding small chicks, relative to 2019 when they all were feeding chicks. In combination, these results suggest birds may have increased their foraging effort during the weak El Niño conditions of 2016 to capture the same amount of prey as birds did in 2019, and to a lesser extent were more likely to opportunistically forage on alternative prey.

Prior studies have examined the diets of Peruvian boobies at various locations and across a range of ENSO conditions. Breeding Peruvian boobies generally have diets consisting primarily of anchoveta [32,36], although prey diversity can vary with latitude [41]. On Lobos de Tierra in northern Peru, Peruvian booby diets contained 93–100% anchoveta by mass in 1996

and 1997 during moderate La Niña and strong El Niño conditions, respectively, with similar regurgitate masses and total prey numbers between years [36]. In addition, birds were completely absent from this island in 1998, during the second year of a strong El Niño event [36]. On Isla Pescadores in central Peru, the diets of Peruvian boobies comprised of 80–93% anchoveta between 2009 to 2013 during neutral and weak to moderate La Niña conditions [32,45].

However, Peruvian boobies further south on Isla Pajaros in north-central Chile exhibit more flexibility in their diet in contrast to northern residents [32,36,41]. For example, diets at this island in 2000 consisted of a mix of anchoveta, king gar, Peruvian silverside, Chilean jack mackerel (*Trachurus murphyi*), Araucanian herring (*Strangomera bentincki*), and squid, while in 2001, anchoveta was completely absent from the diet with Chilean jack mackerel as the dominant (82.3%) prey species in its stead [41]. These latitudinal differences in diet may be a reflection of differences in the seasonality and productivity of upwelling subsystems in the HCS [29]. In addition, it suggests the potential for latitudinal variation in the trophic responses of Peruvian booby populations to ENSO events and the thresholds at which they switch prey and/or abandon breeding.

Sympatric breeding blue-footed and Nazca boobies (*Sula granti*) breeding in the HCS are more flexible in their diets [36]. Unlike Peruvian boobies, these two species did not abandon breeding on Lobos de Tierra during the second year of a strong El Niño event but instead switched prey in response to the low anchoveta availability [36]. In contrast, the distribution of Peruvian boobies in the northern and mid-latitude regions is strongly related to anchoveta distribution [101] and this species exhibits a life history strategy that capitalizes on the boom-bust cycle of anchoveta. As such, given that Peruvian boobies were breeding and had broadly similar diets during the two years of our study, the weak El Niño conditions present in 2016 were likely not strong enough to significantly disrupt anchoveta availability around Isla Guañape Norte.

**Isotopic niche by year.** The bivariate (i.e., $\delta^{13}C$ and $\delta^{15}N$ values) isotopic niche positions of Peruvian boobies on Isla Guañape Norte differed between the weak El Niño conditions of 2016 and the neutral conditions of 2019. In seabirds, $\delta^{13}C$ values commonly serve as a proxy of habitat use, including determining inshore vs. offshore foraging, while $\delta^{15}N$ values are used as a proxy of trophic level [102–106]. As such, one possible interpretation of this result is that the diets and/or foraging habitat of Peruvian boobies on Isla Guañape Norte differed between the two years of our study. However, this interpretation conflicts with the results of our dietary analyses which indicated similar diet composition in both years.

Prior studies have also reported substantial interannual variation in the blood $\delta^{13}C$ and $\delta^{15}N$ values of Peruvian boobies at other breeding locations despite a similar diet composition across years [32,38]. An alternative interpretation of these trends is that the observed interannual variation in the stable isotope values of Peruvian boobies reflect shifts in ecosystem isotopic baselines, not diets or foraging habitat use [105,107]. The HCS is known to experience both latitudinal and temporal variation in isotopic values at the base of the food web [37,38,108], and $\delta^{15}N$ values vary with depth and are related to the OMZ [109]. ENSO events from La Niña to El Niño impact upwelling patterns, the depth of the OMZ, and the dynamics of N-loss processes, which affect the rate of $\delta^{15}N$ fractionation [108]. In addition, shifts in baseline $\delta^{13}C$ value could occur during ENSO events as resident phytoplankton species and size composition change [101,110].

Variation in isotopic values at the base of the food web can then propagate up the HCS food chain to marine predators [38,111]. For example, the blood $\delta^{15}N$ values of Peruvian boobies and Guanay cormorants on Isla Pescadores, Peru were 2–3 ‰ lower during El Niño relative to La Niña conditions [38], presumably due to limited upwelling reducing nitrate supply to surface waters which isotopically depletes particulate organic matter and phytoplankton. In

contrast, we found that the blood $\delta^{15}$N values of Peruvian boobies on Isla Guañape Norte were 0.6 ‰ higher during the weak El Niño conditions present in 2016 relative to the neutral conditions of 2019. Unfortunately, we were not able to sample anchoveta or particulate organic matter for isotopic analyses, nor obtain tracking, diet, and isotope data from all individuals, which might have allowed us to assess if and how isotopic baselines varied between years and to disentangle the trophic vs. baseline effects on the observed isotopic niches of Peruvian boobies on Isla Guañape Norte.

## Conclusion

Despite moderate sexual dimorphism and sex-based differences in body condition, we found little evidence of spatial, behavioral, or trophic niche segregation between male and female Peruvian boobies on Isla Guañape Norte, and few sex-specific differences in diet. One notable exception was prey-size differences, as males captured smaller and more variably-sized prey than females did. While it is not possible to fully tease apart the relative effects of environmental and phenological variation on foraging and trophic metrics examined here, sexual partitioning, or the lack thereof, qualitatively did not appear to vary over the two years of our study. Even so, there was some evidence that both sexes may have increased their foraging efforts during the weak El Niño conditions found in 2016. The lack of sex-specific differences observed in our study may indicate that prey, anchoveta in particular, was abundant enough to not be limiting for the number of adults who opted to breed, precluding the need for sexual segregation as a mechanism to reduce intraspecific competition. However, interannual variation in prey abundance can be extremely high in the HCS [23]. It is possible foraging niche segregation between sexes may occur in response during years of lower prey availability than what was observed in our study. However, it also seems probable that in years of poor prey availability, Peruvian boobies do not rely on foraging niche segregation to reduce intraspecific competition but instead reduce or forgo breeding efforts, or in extremely bad years, simply starve for lack of food [35]. When taking into account the expected increasing frequency of extreme El Niño events due to climate change [112], the sensitivity of the Peruvian booby to anchoveta availability, and the negative relationship between anchoveta availability and El Niño, the relevance of the biomonitoring of this species, for both ecosystem monitoring and conservation purposes, becomes clear. Future studies of Peruvian boobies would benefit from assessing foraging ecology across a wider range of ENSO conditions and colonies to shed light on if and how the foraging and trophic niches of Peruvian boobies varies between sexes relative to food availability across their geographic distribution.

## Supporting information

**S1 Table. Foraging trip metrics of incubating versus chick-rearing Peruvian boobies.** Mean foraging trip metrics and at-sea behaviors ± standard deviations of incubating versus chick-rearing Peruvian boobies GPS tracked on Guañape Norte, Peru in 2016. For statistical tests, we used general linear mixed models specifying individual bird identity as a random effect, and with gamma distributions with log links for trip metrics and beta distributions with logit links for at-sea behaviors. No significant differences were found between groups for any metric. (DOCX)

## Acknowledgements

We thank H. Bennadji, K. Gibson, and A. Stahl for isotope sample processing and analysis, as well as the personnel of Agro Rural, Mr. Moisés Tomairo and Alfredo Flores, for their help and support on Isla Guañape Norte. We are grateful to Yann Tremblay from the French

National Research Institute for Sustainable Development (IRD) who provided the I-gotU GPS loggers in 2016, and to Diego Ardiles, Rachel Quispe, Leonela Valdivia, and Adrián Garaycochea for their assistance on the island in 2016. We thank Diego Acosta for providing his house in Trujillo as headquarters for logistic planning.

## Author contributions

**Conceptualization:** Sara Y. Wang, Carlos Zavalaga, Michael J. Polito.

**Formal analysis:** Sara Y. Wang, Michael J. Polito.

**Funding acquisition:** Sara Y. Wang, Carlos Zavalaga, Michael J. Polito.

**Investigation:** Sara Y. Wang, Carlos Zavalaga, Diego Gonzales-DelCarpio, Cinthia Irigoin-Lovera, Isabella Díaz-Santibañez, Michael J. Polito.

**Methodology:** Sara Y. Wang, Carlos Zavalaga, Michael J. Polito.

**Project administration:** Carlos Zavalaga, Michael J. Polito.

**Resources:** Carlos Zavalaga, Michael J. Polito.

**Supervision:** Carlos Zavalaga, Michael J. Polito.

**Visualization:** Sara Y. Wang, Michael J. Polito.

**Writing – original draft:** Sara Y. Wang.

**Writing – review & editing:** Sara Y. Wang, Carlos Zavalaga, Michael J. Polito.

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
