## [Decision Letter · Decision Letter 0]

18 Jun 2024

PONE-D-24-10208Sexual dimorphism does not translate to foraging or trophic niche partitioning in Peruvian boobies (*Sula variegata* )PLOS ONE

Dear Dr. Wang,

Thank you for submitting your manuscript to PLOS ONE. After careful consideration, we feel that it has merit but does not fully meet PLOS ONE’s publication criteria as it currently stands. Therefore, we invite you to submit a revised version of the manuscript that addresses the points raised during the review process.

When revising your manuscript, pay particular attention to the concern raised by reviewer 2 regarding statistical analyses and sample sizes.

We look forward to receiving your revised manuscript.

Kind regards,

José M. Riascos, Ph.D.

Section Editor

PLOS ONE

Reviewers' comments:

Reviewer's Responses to Questions

**Comments to the Author**

1. Is the manuscript technically sound, and do the data support the conclusions?

Reviewer #1: Yes

Reviewer #2: Partly

2. Has the statistical analysis been performed appropriately and rigorously? 

Reviewer #1: Yes

Reviewer #2: No

3. Have the authors made all data underlying the findings in their manuscript fully available?

Reviewer #1: Yes

Reviewer #2: Yes

4. Is the manuscript presented in an intelligible fashion and written in standard English?

Reviewer #1: Yes

Reviewer #2: Yes

5. Review Comments to the Author

Reviewer #1: The Peruvian booby (Sula variegata, booby hereafter) is a sexually dimorphic seabird living in a highly variable environment under the influence of El Niño Southern Oscillation (ENSO). The authors attempted to determine if there was niche partitioning between sexes by quantifying the foraging behavior and trophic niches of those boobies at Isla Guañape Norte, Peru, by comparing two years with different ENSO conditions. They expected fewer sex-based differences during a “normal” phase (2019) than during El Niño phase (2016, which was nonetheless rather weak). They found that although females were larger and in better body condition than males during both periods, this did not translate into foraging or dietary niche differences. They found that despite there being reverse sexual dimorphism (female larger than male), boobies did not show sex-specific foraging or isotopic niche partitioning and indeed had few differences in diet. The anchoveta pelagic fish (Engraulis ringens) predominated in both male and female diets independently of ENSO phase. They interpreted their results as indicating that there was little intraspecific competition between 2016 and 2019, not leading to the expected trophic niche differentiation.

I find it very valuable that “negative” results such as these are reported: It is –after all– exceptions that make “rules” interesting. Emphasis on confirmatory results is widespread, and for the reason expressed above, I find value in this sobering contribution. That being said, I think that the exceptional nature of the results obtained from the years/ENSO phases contrasted is because the 2016 ENSO was really mild. I would have expected a more likely confirmation for the hypothesis tested if the comparison was made between a “normal” and a strong El Niño phase. Such study awaits to be conducted when the opportunity arises.

The paper is generally carefully written and well analyzed. I do not really have much to suggest about syntax and style. Indeed, the authors are extremely detailed in their presentation, which rather excessively lengthens their manuscript. But I prefer to have more than fewer details provided.

More in a nit-picking mode, I may suggest: (a) Keyword somewhat unnecessarily duplicate what is already in the title. (b) Table 1 and ff: On planet earth, one does not measure mass, only weight. And wing chord, was that flattened or naturally arched? Rather that SD, one should report 2SE, which immediately gives you an idea of the 95% confidence interval for the mean. (c) In Table 3: I cannot find the asterisks announced in the legend.

Reviewer #2: Sexual dimorphism does not translate to foraging or trophic niche partitioning in Peruvian boobies (Sula variegata)

Overall appreciation

This manuscript assessed the sexual and interannual differences (between two years) on the foraging distribution, at-sea behaviour, isotopic niche (δ13C-δ15N), and diet composition (through regurgitate analysis) of Peruvian boobies, during a year of neutral vs. weak El Niño conditions. Overall, the manuscript is well-written and provides new insights into the ecology of Peruvian boobies during the breeding season. However, I found some parts unclear throughout the manuscript. My comments offer suggestions for improvement, but in other cases I raise questions or suggest additional justification is needed. I hope my comments are clear, potentially revealing aspects of the work that I may have misunderstood and are helpful for improving the manuscript.

Major concerns:

1. The premise that weak El Niño conditions would trigger differences in foraging and trophic ecology lacks sufficient support from the literature. I believe this premise is not strong enough to justify assessing interannual differences. The lower sample size in 2016 (n = 10) compared to 2019 (n = 32) adds uncertainty to the interannual assessment, especially when comparing sexes within each year (7 females vs. 3 males in 2016; 20 females vs. 12 males in 2019). There are also some misconceptions in the methodology and some biased interpretations of the results.

2. I strongly recommend the authors revisit some crucial aspects of the statistical design, particularly in the analysis of GPS data and regurgitate data, as well as other aspects related to tracking data representativeness and calculation of smoothing parameters.

3. I am not entirely convinced of the importance of analysing and discussing the degree of sexual dimorphism in this study. In fact, there is no linear pattern found for this topic. Instead, I would expect a discussion focused on the impact or influence of sexual dimorphism on adult foraging metrics, at-sea behaviour and spatial segregation, diet composition, and isotopic niche, rather than just a comparison with the degree of sexual dimorphism in other species. While it is a good premise to discuss your results, it should not be discussed exhaustively.

4. The discussion is not structured in an easily understandable way. I suggest dividing the discussion into two sections: one for sexual differences and another for interannual differences (even though I have significant doubts about the representativeness of GPS and isotopic data in 2016). This should simplify the discussion and avoid the use of many subsections that, in my opinion, do not warrant such emphasis.

Specific comments

Title: amend “translate to foraging” to “translate into foraging”.

Abstract

Line 2: amend “marine birds” to “seabirds”.

Line 14: I would amend to ‘sexual dimorphism’, removing the ‘reverse’. If the males were larger than the females these would be expected anyway.

Line 19-20: And when they are unfavourable too, correct? According to the results, authors have stated there were no interannual differences, so I would rather suggest that oceanographic conditions were not that strong to influence prey intake, or the type of prey captured by males and/or females in both years. In fact, it suggests that both prey and predators are adapted to these conditions in the Humboldt Current System.

Keywords: The use of “niche partitioning”, “Peruvian booby”, and “sexual dimorphism” as keywords will not increase the range of readers since they are already mentioned in the title. Try to use different keywords, like: Sulidae, resource partitioning, reverse sexual dimorphism (RSD).

Introduction

Line 12-13: I suggest changing to “sexual dimorphism” or “size-related sexual differences”.

Line 62-63: This sentence is redundant. In the two previous sentences authors have already reported the “slight” differences on the foraging metrics.

Line 66: The references are not properly formatted.

Line 72-73: I would say this is not an expectation concerning the already known reverse sexual dimorphism. What authors could expect is a higher or lower size dimorphism index (SDI).

Line 76-79: I fully agree with this, however it felt incomplete. I would suggest attributing specific expected patterns to males and females regarding foraging metrics/characteristics, behaviour, isotopic niche, prey intake, mass, or size. Despite previous studies reporting low evidence of sex-based differences in tropical environments, authors can infer expected differences in this study, particularly under less favourable conditions. Indeed, I would expect a priori a competitive advantage for females due to their larger body size. For instance, Almeida et al., 2021 (already cited in this manuscript) observed sexual segregation in brown boobies only during a period of co-existence with another breeding sulid, indicating a period of higher adult density within the foraging range.

Materials and Methods

Line 105: Chicks or eggs.

Line 109: The reference is not properly formatted.

Line 116: How were the feathers marked?

Line 124-125: Perhaps I have missed something, but it looks like the tag was only deployed on the adults for a few hours.

Line 130-132: If I am not wrong, the blood cells are those who have a turnover rate of 28 days for δ13C and δ15N values. The turnover rate in whole blood is larger than that, even though comparable isotopically speaking according to Cherel et al. (2005).

Cherel, Y., Hobson, K.A., Bailleul, F., Groscolas, R., 2005. Nutrition, physiology, and stable isotopes: new information from fasting and molting penguins. Ecology 86, 2881–2888.

Line 154-156: I would rather suggest the use of a radius (perhaps of 200 m). It is more uniform.

Line 159-161: This is why I prefer to use a radius. Sometimes, boobies can interact when at the colony or when at the arriving/landing which may disturb each other and result in brief flights. According to the authors’ data filtering method, the occurrence of false foraging trips seems likely.

Line 184-185: Perhaps some literature support in here or at least a brief explanation of why pooling these two categories together.

Line 192-193: The smoothing parameter was calculated for each group? If yes, 3 males in 2016 is a very low number of individuals to compute such parameter. Please clarify.

Line 209: Aside from all these metrics, an important request when working with tracking data at the population level is the representativeness of the data. Have authors checked for it? A good way to verify the representativeness of the foraging areas is to follow the method described here: Lascelles, BG et al. Applying global criteria to tracking data to define important areas for marine conservation. Divers Distrib 22: 422−431 (2016), and applied here: Cecere, JG et al. Spatial segregation of home ranges between neighbouring colonies in a diurnal raptor. Sci Rep 8: 11762 (2018); here: Weimerskirch, H. et al. At-sea movements of wedge-tailed shearwaters during and outside the breeding season from four colonies in New Caledonia. Mar. Ecol. Prog. Ser. 663, 225–238 (2020); and here: dos Santos, I. et al. Sexual segregation in the foraging distribution, behaviour, and trophic niche of the endemic Boyd’s shearwater (Puffinus lherminieri boydi). Mar Biol 169 (2022).

Line 214-215: Have authors tested for multicollinearity? If not, it should be done because it is not necessary to test collinear variables.

Line 216-217: My only concern with this design is the lack of males successfully tracked in 2016. I truly believe that this may have influenced the results.

Line 218: Have authors checked the residual deviance of the random factor, the residuals of the model through qqplots (normality) and homogeneity of variances (to detect potential biases)?

Line 219: Were these data transformed? I am asking because authors have mentioned the log-transformation of the foraging metrics but not the behaviours. Please clarify.

Line 220: Any p-adjustments for multiple modelling for the using of the same dataset?

Line 231: What does this mean precisely? Percent of prey by mass of regurgitate? Please clarify.

Line 236-240: My concern here is that some of these metrics may be correlated as in foraging trip parameters. For instance, higher the number of prey items, higher the total regurgitate mass; or higher the individual prey size, lower the number of prey items. So, these tests are not independent, and this should be considered.

Line 251: Since no context was given about stable isotopes and their usefulness in ecology and food web studies, it would be informative to add in here a brief explanation of how carbon and nitrogen can be informative in diet studies.

Line 253: Just a minor correction, the only unit that does not need a space between the number and the unit is the percentage (%). For all the other units authors should use the space between.

Results

Line 310: Two decimal cases are sufficient for the tests and three decimal cases for the p-values.

Line 322-326: What is the reason for testing annual differences on morphometrics? With exception of body mass that could indicate a lower overall condition, it would already be expected that females were larger than males.

Line 334-340: I would resume this section in: “There were no annual either sex differences on any of the foraging trip metrics (minimum F < F < maximum F; p-value > minimum p-value). Also, as mentioned before, these variables are probably collinear. If so, authors only need to test the effect of one out of three predictors.

Line 344-347: I am not aware of the journal limits of characters in the legends of tables and figures, but if it is possible, include this part in the main legend as well. The same for the other tables.

Line 349: Testing all these proportions, authors could not guarantee the independence of data. These variables are highly dependent from each other since they correspond to the proportion of time spent in each one of the four categories. I would rather test the foraging behaviour.

Line 425: Is there a reason for table 3 comes after table 4 in the manuscript?

Line 426: Just a minor correction, the delta sign (as alpha, beta, gamma) must be in italics.

Line 430: Just a minor correction, the symbol used for carbon values is not a minus sign as in maths, but a “–“.

Line 451-460: No need to present both sexes in relation to one another. Just present females realtive to males or males relative to females. Otherwise, it is redundant.

Figure 3: This figure needs considerable improvements. The legend of the figure is overlayed with the fourth plot. The symbol per mil could be represented instead of “per mil” in the xx’ axis title. The use of more contrasting colours and the use of a colour-blind friendly pallette are recommended.

Discussion

Line 469-470: I would remove this sentence since this was not a main finding of your study. This was already reported.

Linbe 476-477: This was one of the most interesting results and it was not given much relevance.

Line 479: “both sexes spent proportionally less time foraging in 2019 than in 2016.” This section looks disconnected from the sentence. Perhaps starting a new sentence when shifting the subject to foraging activity.

Line 483-511: This would be useful on explaining the differences, or in this case the absence of differences, on foraging metrics, behaviour, and isotopic niche instead of describing something that was already known amnd compare with other populations or even species. Do not get me wrong, it is a good premise to discuss your results but not to be described exhaustively.

Line 561: The reference is not properly formatted.

Line 573: Authors can also add the ref 52 (Almeida et al. 2021) in red-footed and brown boobies.

Line 584-587: Also, it is possible that: 1) El Niño weak conditions in 2016 were not enough (similar prey abundance and availability to adults) to trigger sexual segregation on diet composition and/or on foraging behaviour while at sea; 2) The low sample size in 2016 could have masked some potential sex-related differences on isotopic niches.

Line 597-598: It could have extended their foraging ranges, travel distances, and trip duration. Interestingly, this pattern was only detected for females, at least concerning the home range areas (95% KUD).

Line 614-616: Out of curiosity, are there some strong evidence in the literature of different parental roles in this species?

Line 618-620: Indeed this can happen. However, with weak El Niño conditions it would not be highly expected. Moreover, the prey mass did not change from 2016 to 2019, it has only varied between sexes, mostly due (I would say) to size-related sex differences.

Line 623: Please amend “dispersed” to “sparse”.

Line 623-624: What is the difference between “laterally” and “horizontally”. Please clarify.

Line 626: Please amend “labor” to “parental duties” or just “duties”.

Line 627: Please amend “prey scarce years” to “years of lower prey availability” or “years of prey scarcity”.

Line 638: Too much ‘howevers’. Avoid the over use of adverbs.

Line 675-677: Peruvian boobies are quite specialist then. This scenario is similar to the extreme breedigng failure of common guillemot on the coast of Britain in 2004, which was due to the demographic collapse of sandeels (Wanless et al., 2005).

Wanless, S., Harris, M. P., Redman, P., & Speakman, J. R. (2005). Low energy values of fish as a probable cause of a major seabird breeding failure in the North Sea. Marine Ecology Progress Series, 294, 1-8.

Line 713: The shift of isotopic baselines of carbon and nitrogen is not the only factor to mention here. In fact, the absence of support of the different approaches used in this study could be due to three factors (in separate or in combination):

1) The individuals sampled for diet, isotopes analysis, and those tracked with GPS were not the same. Thus, inter-individual differences were not possible to take into consideration;

2) The representativeness of this Peruvian’s booby population was not guaranteed by the number of individuals tracked with GPS devices;

3) the large differences between the sample sizes of 2016 and 2019 may have also biased the results, hiding or not possible interannual patterns.

6. PLOS authors have the option to publish the peer review history of their article (what does this mean? ). If published, this will include your full peer review and any attached files.

**Do you want your identity to be public for this peer review?** For information about this choice, including consent withdrawal, please see our Privacy Policy .

Reviewer #1: **Yes: ** fabian m. Jaksic

Reviewer #2: **Yes: ** Ivo dos Santos

---

## [Author Response · Author response to Decision Letter 1]

3 Sep 2024

See attached Response to Reviewers document.

---

## [Decision Letter · Decision Letter 1]

11 Dec 2024

PONE-D-24-10208R1Sexual dimorphism does not translate into foraging or trophic niche partitioning in Peruvian boobies (*Sula variegata* )PLOS ONE

Dear Dr. Wang,

Thank you for submitting your manuscript to PLOS ONE. After careful consideration, we feel that it has merit but does not fully meet PLOS ONE’s publication criteria as it currently stands. Therefore, we invite you to submit a revised version of the manuscript that addresses the points raised during the review process.

In reading the original reviewer reports, as well as the most recent reviewer report (which recommends rejection), we are not convinced that this paper can be published in its current form. One original reviewer had already identified the flaws that reviewer #3 now also highlights, which are not only the low sample size in 2016, but also a completely different breeding stage in 2016 versus 2019. The breeding stage difference could explain any inter annual differences found, so it becomes virtually impossible to disentangle whether any differences found are due to weather/oceanographic differences (as the paper wishes to address) versus breeding stage differences. We encourage the authors to focus solely on analyzing intersexual differences for the 2019 dataset. Authors could opt to report the 2016 data (since they collected it and it is informative) but refrain from statistically analyzing inter annual differences. We also found some inconsistency in the reported sample sizes. For example, within the methods, it states that, in 2016, 14 females and 4 males were "targeted for capture", but then in table 4 it is reported that blood samples were taken from 14 females and 9 males - which one is correct? We attempted to understand this by looking at the datasets, but those were not provided. Please clearly report the sample sizes for each sample type/sex/year.

We look forward to receiving your revised manuscript.

Kind regards,

Patricia C. Lopes

Academic Editor

PLOS ONE

Reviewers' comments:

Reviewer's Responses to Questions

**Comments to the Author**

1. If the authors have adequately addressed your comments raised in a previous round of review and you feel that this manuscript is now acceptable for publication, you may indicate that here to bypass the “Comments to the Author” section, enter your conflict of interest statement in the “Confidential to Editor” section, and submit your "Accept" recommendation.

Reviewer #2: All comments have been addressed

Reviewer #3: All comments have been addressed

2. Is the manuscript technically sound, and do the data support the conclusions?

Reviewer #2: Yes

Reviewer #3: No

3. Has the statistical analysis been performed appropriately and rigorously? 

Reviewer #2: Yes

Reviewer #3: No

4. Have the authors made all data underlying the findings in their manuscript fully available?

Reviewer #2: Yes

Reviewer #3: Yes

5. Is the manuscript presented in an intelligible fashion and written in standard English?

Reviewer #2: Yes

Reviewer #3: Yes

6. Review Comments to the Author

Reviewer #2: Overall, I am truly satisfied with this revised version of the manuscript entitled “Sexual dimorphism does not translate into foraging or trophic niche partitioning in Peruvian boobies (Sula variegata)”. The authors have responded fully to the comments made during the review, and I commend the authors for completing such an extensive revision. The changes made in response to the reviewers' comments have significantly improved the quality of the manuscript, which is now almost ready for publication. I have only a few minor comments to make before acceptance.

Minor comments:

Line 100-112: I appreciate this clarification. Now it is much more perceptible that the proportion of the nests tracked during this study was relevant in both years.

Line 279: Remove the comma in “homogeneity of variance, and is”.

Line 552: Amend “among” to “between”.

Line 569: Remove the “of” in “asymptotic relationship of between”.

Line 575: Remove the duplicate “in”.

Line 586: Suggest remove “also”.

Line 586: Suggest remove “than males did on average”. It is redundant since you are always comparing males and females.

Line 589: Remove “for us”.

Line 593: Replace “or” by “nor”.

Line 596: Amend “mass” to “size”.

Line 609: Suggest remove “we observed”.

Line 738: “tissue δ13C value”. No need for “tissue” in here.

Line 765: Remove “us”.

Line 765: Remove “and studies”. No need for it I guess.

Line 775-776: “there was some evidence to suggest”. No need to be that cautious here. I would remove “to suggest” from the sentence.

Line 777: I think “present in 2016” could be removed. Authors have already categorised the years of respective weak El Niño and La Niña years above.

Line 778: Remove “in each year”. Anchoveta was abundant across the years.

Line 781: “Interannual variation in prey abundance can be extremely high in the HCS (23) and so it is possible foraging niche segregation between sexes may occur in response during years of lower prey availability than what was observed in our study.” I would divide in two separate sentences. Like, “Interannual variation in prey abundance can be extremely high in the HCS (23). Thus, sex-related foraging niche segregation may occur during years of lower prey availability than what was observed in our study.”

Reviewer #3: This study investigated in the Peruvian booby, sexual differences in foraging and trophic niches in two years with different oceanographic conditions. The study was carried out during December 2016 considered a weak El Niño year, and during November 2019 considered a year with neutral conditions. For these purposes, they compared morphometrics, foraging behavior from GPS tracking, diets via regurgitates, and isotopic niches of males and females and between years. The study is well presented, and the combination of methods and statistical analyses used are appropriate to address this type of question. Unfortunately, the sample size for 2016 is too small to test the main question of the study: sexual differences in foraging and trophic niches in different oceanographic conditions. During 2016, 7 females and 3 males were tracked to record foraging behavior, and blood sampled for isotopic analyses. With this sample size, the reliability of estimates to test the interaction between year*sex is very poor. Additionally, for the analyses of diet via regurgitates, the sex of the adult was not recorded in 2016, so again, the authors are unable to test the relevant interaction.

Additional comments:

Line 32. I think the sentence is misleading. Peruvian boobies do not aggregate in dense colonies during breeding to feed… on a bounty of anchoveta. It is more likely the other way around, because they can feed on a bounty of anchoveta that they aggregate in dense colonies for reproduction.…

Lines 148-149. Only one day of tracking was recorded per bird, which is insufficient to describe the foraging behavior of an individual.

Lines 100-103. Description of the oceanographic conditions for the first sampling period of the study considered a weak El Niño is confusing because, according to the authors, there was a “….prolonged moderate-to-weak El Niño that lasted most of 2016 (February–July) (27,45). The 2016 sampling period was followed by another moderate El Niño (February–April 2017).”. Hence, it is not clear whether the sampling period (December 2016) was under El Niño conditions or between two El Niño’s.

Lines 117-120. Focal adults studied during the two years were in different breeding stages, which may influence foraging behavior of parents. In 2016, focal adults were incubating or rearing recently hatched chicks, whereas in 2019, focal adults were feeding chicks between 3-8 weeks old. The frequency of feeding chicks from ages that range almost 2 months varies considerably, and these changes in chick requirements/feeding frequency may influence some components of the foraging behavior of parents (e.g. duration of foraging trips or the distance to the foraging site). I think the authors should consider this topic when interpreting the results, as it may provide another possible source of variation.

Line 270. P-value is missing.

7. PLOS authors have the option to publish the peer review history of their article (what does this mean? ). If published, this will include your full peer review and any attached files.

**Do you want your identity to be public for this peer review?** For information about this choice, including consent withdrawal, please see our Privacy Policy .

Reviewer #2: **Yes: ** Ivo dos Santos

Reviewer #3: No

---

## [Author Response · Author response to Decision Letter 2]

24 Jan 2025

Text below also included in attached "Response to Reviewers" document.

RESPONSE TO REVIEWERS

Wang et al. - PONE-D-24-10208R1

Academic Editor

1. One original reviewer had already identified the flaws that reviewer #3 now also highlights, which are not only the low sample size in 2016, but also a completely different breeding stage in 2016 versus 2019. The breeding stage difference could explain any inter annual differences found, so it becomes virtually impossible to disentangle whether any differences found are due to weather/oceanographic differences (as the paper wishes to address) versus breeding stage differences. We encourage the authors to focus solely on analyzing intersexual differences for the 2019 dataset. Authors could opt to report the 2016 data (since they collected it and it is informative) but refrain from statistically analyzing inter annual differences.

o Response: We appreciate this constructive feedback and guidance on how to move forward. Following your guidance, we have revised our manuscript to focus primarily on intersexual differences as suggested, while conservatively discussing data from 2016 and interannual variation where sample sizes allow given the rarity and value of data from ENSO years when breeding numbers are low. To achieve this, significant portions of the introduction, methods, statistical analysis and discussion have been revised and updated to reflect this restructuring of the manuscript. Specifically, we focus the statistical analysis of GPS tracking data to 2019, and only qualitatively report data and comparisons that involve 2016. While we have retained statistical comparisons of the regurgitate and stable isotope datasets between years and sexes (due to their higher 2016 samples sizes relative to GPS tracking), we have significantly revised the introduction and discussion to be much more conservative in respect to interpreting any observed inter-annual trends. Specifically, we removed all hypotheses relating to environmental differences, and instead simply test for differences and conservatively discuss how variation in environmental conditions, differing parental roles, and/or breeding phenology between years might have influenced our results. We also explicitly state in our conclusions that conclusively teasing apart the relative influences of environmental and phenological variation on inter-annual trends is not possible given the available data. Thank you for this guidance. We feel that these major revisions, along with providing specific clarity on the exact sample sizes available for each dataset (see below), and a point-by-point response to the reviewers’ comments have addressed the stated concerns and improved the final manuscript.

2. We also found some inconsistency in the reported sample sizes. For example, within the methods, it states that, in 2016, 14 females and 4 males were "targeted for capture", but then in table 4 it is reported that blood samples were taken from 14 females and 9 males - which one is correct? We attempted to understand this by looking at the datasets, but those were not provided. Please clearly report the sample sizes for each sample type/sex/year.

o Response: In 2016, 14 females and 9 males total were captured and sampled for blood, but of these individuals only 7 females and 3 males were successfully tracked via GPS. We have clarified the sample sizes in the manuscript accordingly and ensured the sample sizes in the subsections for each method are clearly stated.

Reviewer #2

Overall, I am truly satisfied with this revised version of the manuscript entitled “Sexual dimorphism does not translate into foraging or trophic niche partitioning in Peruvian boobies (Sula variegata)”. The authors have responded fully to the comments made during the review, and I commend the authors for completing such an extensive revision. The changes made in response to the reviewers' comments have significantly improved the quality of the manuscript, which is now almost ready for publication. I have only a few minor comments to make before acceptance.

• Response: Thank you for re-reviewing our manuscript and for your positive feedback in response to the initial revision. While we have revised the manuscript once again in response to editorial guidance and constructive feedback from a new reviewer, we have sought to retain the improvements made during the prior round of revisions.

Minor comments:

• Line 100-112: I appreciate this clarification. Now it is much more perceptible that the proportion of the nests tracked during this study was relevant in both years.

o Response: We appreciate your previous feedback that helped to bring forward this point and improve our methods section.

• Line 279: Remove the comma in “homogeneity of variance, and is”.

o Response: Revised as suggested.

• Line 552: Amend “among” to “between”.

o Response: Revised as suggested.

• Line 569: Remove the “of” in “asymptotic relationship of between”.

o Response: Revised as suggested.

• Line 575: Remove the duplicate “in”.

o Response: Revised as suggested.

• Line 586: Suggest remove “also”.

o Response: Revised as suggested.

• Line 586: Suggest remove “than males did on average”. It is redundant since you are always comparing males and females.

o Response: Revised as suggested. We kept the descriptor “on average” and moved it to the beginning of “captured larger anchoveta”. Although we detail the distributions of sizes captured by each sex in the following sentence, we feel it is important to keep this descriptor for accuracy, as some female-captured prey had smaller total lengths than some male-captured prey.

• Line 589: Remove “for us”.

o Response: Revised as suggested.

• Line 593: Replace “or” by “nor”.

o Response: We have kept this as is. “Or” is grammatically correct in this sentence.

• Line 596: Amend “mass” to “size”.

o Response: We have revised for clarity by specifying “regurgitate mass” and “prey size”, as we are referring to both of these measurements.

• Line 609: Suggest remove “we observed”.

o Response: Revised as suggested.

• Line 738: “tissue δ13C value”. No need for “tissue” in here.

o Response: Revised as suggested.

• Line 765: Remove “us”.

o Response: We have kept the sentence as is in order to preserve the grammatical structure of the following clauses.

• Line 765: Remove “and studies”. No need for it I guess.

o Response: We agree. Revised as suggested.

• Line 775-776: “there was some evidence to suggest”. No need to be that cautious here. I would remove “to suggest” from the sentence.

o Response: Revised as suggested.

• Line 777: I think “present in 2016” could be removed. Authors have already categorised the years of respective weak El Niño and La Niña years above.

o Response: Revised as suggested.

• Line 778: Remove “in each year”. Anchoveta was abundant across the years.

o Response: Revised as suggested.

• Line 781: “Interannual variation in prey abundance can be extremely high in the HCS (23) and so it is possible foraging niche segregation between sexes may occur in response during years of lower prey availability than what was observed in our study.” I would divide in two separate sentences. Like, “Interannual variation in prey abundance can be extremely high in the HCS (23). Thus, sex-related foraging niche segregation may occur during years of lower prey availability than what was observed in our study.”

o Response: We have split this sentence into two.

Reviewer #3

This study investigated in the Peruvian booby, sexual differences in foraging and trophic niches in two years with different oceanographic conditions. The study was carried out during December 2016 considered a weak El Niño year, and during November 2019 considered a year with neutral conditions. For these purposes, they compared morphometrics, foraging behavior from GPS tracking, diets via regurgitates, and isotopic niches of males and females and between years. The study is well presented, and the combination of methods and statistical analyses used are appropriate to address this type of question. Unfortunately, the sample size for 2016 is too small to test the main question of the study: sexual differences in foraging and trophic niches in different oceanographic conditions. During 2016, 7 females and 3 males were tracked to record foraging behavior, and blood sampled for isotopic analyses. With this sample size, the reliability of estimates to test the interaction between year*sex is very poor. Additionally, for the analyses of diet via regurgitates, the sex of the adult was not recorded in 2016, so again, the authors are unable to test the relevant interaction.

• Response: Thank you for reviewing our manuscript and for your constructive feedback. In response to the comments and concerns raised we have revised the manuscript following the specific guidance provide by the handling editor. As noted above, we focus our statistical analysis of GPS data on 2019 given the low sample sizes in 2016. We also better clarify the higher number of birds sampled for regurgitation and blood isotopes (relative to GPS tracking) and remove hypotheses related to ENSO and interannual variation. Finaly, where sample sizes allow, we provide a more conservative interpretation of interannual trends that highlight the inherent challenge of conclusively teasing apart the relative influences of environmental and phenological variation.

Additional comments:

• Line 32. I think the sentence is misleading. Peruvian boobies do not aggregate in dense colonies during breeding to feed… on a bounty of anchoveta. It is more likely the other way around, because they can feed on a bounty of anchoveta that they aggregate in dense colonies for reproduction.…

o Response: We have revised this sentence to better reflect this point.

• Lines 148-149. Only one day of tracking was recorded per bird, which is insufficient to describe the foraging behavior of an individual.

o Response: We deployed tags for a shorter period of time for several reasons. First, due to the limited availability of biologgers, we decided to deploy them for just a period of one day in order to gather more data from different individuals to avoid pseudoreplication. Second, the device battery life was also fairly short, since these were high-resolution data loggers. And lastly, we decided on shorter deployment times to minimize any potential handicapping effect on the birds that can occur with longer deployment times. This approach allowed us to capture a wider trend in the population versus focusing on repeated measures of individual behaviors.

• Lines 100-103. Description of the oceanographic conditions for the first sampling period of the study considered a weak El Niño is confusing because, according to the authors, there was a “….prolonged moderate-to-weak El Niño that lasted most of 2016 (February–July) (27,45). The 2016 sampling period was followed by another moderate El Niño (February–April 2017).”. Hence, it is not clear whether the sampling period (December 2016) was under El Niño conditions or between two El Niño’s.

o Response: We have revised the first sentence of the paragraph (now Line 96) to better clarify that sampling occurred during El Niño conditions. We hope that the context, that this was a weak El Niño further sandwiched between two prolonged periods of strong/moderate El Niño conditions, is clear.

• Lines 117-120. Focal adults studied during the two years were in different breeding stages, which may influence foraging behavior of parents. In 2016, focal adults were incubating or rearing recently hatched chicks, whereas in 2019, focal adults were feeding chicks between 3-8 weeks old. The frequency of feeding chicks from ages that range almost 2 months varies considerably, and these changes in chick requirements/feeding frequency may influence some components of the foraging behavior of parents (e.g. duration of foraging trips or the distance to the foraging site). I think the authors should consider this topic when interpreting the results, as it may provide another possible source of variation.

o Response: Thank you for this comment. Our original manuscript discussed the possible effects and biases of differences in breeding phenology – though given this comment and the guidance from the handling editor we agree that the manuscript would be improved by a more conservative interpretation of inter-annual trends. As such, we have removed the specific hypotheses related to ENSO and interannual variation, and while we have retained inter-annual comparisons where sample sizes allow, we now provide a more conservative interpretation of our results that better highlight the challenge of conclusively teasing apart the relative influences of environmental and phenological variation. For example, see Lines 71-74, 150-154, 251-253, 530-535, 637-345, 702-705, and 777-780 among others in the revised manuscript for examples of these changes.

• Line 270. P-value is missing.

o Response: We have corrected the stated p-value.

---

## [Decision Letter · Decision Letter 2]

14 Feb 2025

Sexual dimorphism does not translate into foraging or trophic niche partitioning in Peruvian boobies (*Sula variegata* )

PONE-D-24-10208R2

Dear Dr. Wang,

We’re pleased to inform you that your manuscript has been judged scientifically suitable for publication and will be formally accepted for publication once it meets all outstanding technical requirements and final suggestions by the reviewer.

Kind regards,

Claudia Mettke-Hofmann, PhD

Section Editor

PLOS ONE

Additional Editor Comments (optional):

Comments by the Academic Editor:

The manuscript has been further improved and is now in the range to be accepted. Please address the comments made by the reviewer and the Academic Editor.

Reviewers' comments:

Reviewer's Responses to Questions

**Comments to the Author**

1. If the authors have adequately addressed your comments raised in a previous round of review and you feel that this manuscript is now acceptable for publication, you may indicate that here to bypass the “Comments to the Author” section, enter your conflict of interest statement in the “Confidential to Editor” section, and submit your "Accept" recommendation.

Reviewer #2: All comments have been addressed

2. Is the manuscript technically sound, and do the data support the conclusions?

Reviewer #2: Yes

3. Has the statistical analysis been performed appropriately and rigorously? 

Reviewer #2: Yes

4. Have the authors made all data underlying the findings in their manuscript fully available?

Reviewer #2: Yes

5. Is the manuscript presented in an intelligible fashion and written in standard English?

Reviewer #2: Yes

6. Review Comments to the Author

Reviewer #2: Overall appreciation

Overall, I am truly satisfied with this third revised version of this manuscript. The changes made especially in response to the new reviewer have significantly improved the quality and organization of the manuscript. I have only a few minor comments to make before acceptance.

Minor comments

Discussion

L550-552: “These prior studies have reported similar trip distances, trip durations, and maximum foraging distances from colony between sexes and moderate to high overlap (20–74%) of foraging areas (26,42).” First authors state that in previous studies maximum foraging distances have been reported to be similar between sexes. However, later in L554, it is detailed that in Weimerskirch et al., (42), females were reported to have slightly greater maximum foraging distances than males. It sounds a little bit contradictory. Is this “slightly different” statistically different? Or is it just different on their absolute value? Please clarify.

L750: Please remove the repeated “is that”

Conclusion

L773-793: Perhaps it would be good to add a sentence concerning the climate changes scenario which could make El Niño events even stronger and more frequent. Authors have already stated that Peruvian boobies are highly sensitive to the fluctuations of Anchoveta availability, however, I believe it would reinforce the importance of biomonitoring of this species breeding in the HCS.

7. PLOS authors have the option to publish the peer review history of their article (what does this mean? ). If published, this will include your full peer review and any attached files.

**Do you want your identity to be public for this peer review?** For information about this choice, including consent withdrawal, please see our Privacy Policy .

Reviewer #2: **Yes: ** Ivo dos Santos

---

## [Editor Report · Acceptance letter]

PONE-D-24-10208R2

PLOS ONE

Dear Dr. Wang,

I'm pleased to inform you that your manuscript has been deemed suitable for publication in PLOS ONE. Congratulations! Your manuscript is now being handed over to our production team.

Kind regards,

on behalf of

Prof Claudia Mettke-Hofmann

Section Editor

PLOS ONE